# Nationwide incidence of sarcomas and connective tissue tumors of intermediate malignancy over four years using an expert pathology review network

**Gonzague de Pinieux**[1☯], **Marie Karanian**[2☯], **Francois Le Loarer**[3☯], **Sophie Le Guellec**[4], **Sylvie Chabaud**[2], **Philippe Terrier**[5], **Corinne Bouvier**[6], **Maxime Batistella**[7], **Agnès Neuville**[3], **Yves-Marie Robin**[8], **Jean-Francois Emile**[9], **Anne Moreau**[10], **Frederique Larousserie**[11], **Agnes Leroux**[12], **Nathalie Stock**[13], **Marick Lae**[13,14], **Francoise Collin**[15], **Nicolas Weinbreck**[16], **Sebastien Aubert**[8], **Florence Mishellany**[17], **Celine Charon-Barra**[15], **Sabrina Croce**[3], **Laurent Doucet**[18], **Isabelle Quintin-Rouet**[18], **Marie-Christine Chateau**[19], **Celine Bazille**[20], **Isabelle Valo**[21], **Bruno Chetaille**[16], **Nicolas Ortonne**[22], **Anne Brouchet**[4], **Philippe Rochaix**[4], **Anne Demuret**[1], **Jean-Pierre Ghnassia**[23], **Lenaig Mescam**[24], **Nicolas Macagno**[6], **Isabelle Birtwisle-Peyrottes**[25], **Christophe Delfour**[19], **Emilie Angot**[13], **Isabelle Pommepuy**[26], **Dominique Ranchere**[2], **Claire Chemin-Airiau**[2], **Myriam Jean-Denis**[2], **Yohan Fayet**[2], **Jean-Baptiste Courrèges**[3], **Nouria Mesli**[3], **Juliane Berchoud**[10], **Maud Toulmonde**[3], **Antoine Italiano**[3], **Axel Le Cesne**[2], **Nicolas Penel**[8], **Francoise Ducimetiere**[2], **Francois Gouin**[2‡], **Jean-Michel Coindre**[3‡], **Jean-Yves Blay**[2,27,28‡]*, on behalf of the NetSarc/RePPS/ResSos and French Sarcoma Group-Groupe d'Etude des Tumeurs Osseuses (GSF-GETO) networks[¶]

1 Department of pathology, CHU de Tours, Tours, France, 2 Department of Biopathology, Centre Léon Bérard, Lyon, France, 3 Department of Biopathology, Institut Bergonié, Bordeaux, France, 4 Department of Biopathology, Institut Claudius Regaud et Institut Universitaire du Cancer de Toulouse—Oncopôle, Toulouse, France, 5 Department of Biopathology, Gustave Roussy, Villejuif, France, 6 Department of pathology, La Timone University Hospital, Marseille, France, 7 Pathology Department, Saint-Louis Hospital, AP-HP, Université de Paris, Paris, France, 8 Pôle de Biologie-Pathologie-Génétique Centre Oscar Lambret, & Institut de Pathologie entre Oscar Lambret & CHU Lille, Lille, France, 9 Department of Biopathology, Hopital Ambroise Paré, Boulogne, France, 10 Department of Pathology, Department of Orthopedy CHU Nantes, Nantes, France, 11 Department of Biopathology, Hôpital Cochin-Saint-Vincent de Paul, Paris, France, 12 Department of Biopathology, Institut de Cancérologie de Lorraine—Alexis Vautrin, Vandoeuvre-lès-Nancy, France, 13 Department of Biopathology, Eugene Marquis Comprehensive Cancer Center & CHU Rennes, Rennes, France, 14 Department of Biopathology, Institut Curie, Paris, France, 15 Department of Biopathology, Centre Georges François Leclerc, Dijon, France, 16 Medipath, Frejus, France, 17 Department of Biopathology, Centre Jean Perrin, Clermont-Ferrand, France, 18 Department of pathology, CHRU Brest, Brest, France, 19 Department of Biopathology, Institut de Cancérologie de Montpellier & CHU Montpellier, Montpellier, France, 20 Department of Biopathology, Centre Francois Baclesse, Caen, France, 21 Department of Pathology, Institut de Cancerologie de l'Ouest, Angers, France, 22 Department of Biopathology, Hopital Henri Mondor, Creteil, France, 23 Department of Biopathology, Centre Paul Strauss, Strasbourg, France, 24 Department of Biopathology, Institut Paoli Calmettes, Marseille, France, 25 Department of Biopathology, Centre Antoine-Lacassagne, Nice, France, 26 Department of Biopathology, CHU Limoges, Limoges, France, 27 Department of Medicine of Centre Leon Berard, University Claude Bernard Lyon I, Lyon, France, 28 Headquarters, Unicancer, Paris, France

☯ These authors contributed equally to this work.
‡ These authors also contributed equally to this work.
¶ Membership of NetSarc/RePPS/ResSos and French Sarcoma Group- Groupe d'Etude des Tumeurs Osseuses (GSF-GETO) networks is provided in the S1 Table.
* jean-yves.blay@lyon.unicancer.fr

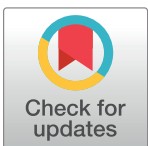

**Data Availability Statement:** All data is in the tables and supplemental documents.

**Funding:** This work was supported by Institut National Du Cancer (FR) NETSARC+, Institut National Du Cancer (FR) RREPS, Institut National Du Cancer (FR) RESOS, Institut National Du Cancer (FR) NETSARC, Institut National Du Cancer (FR) LYriCAN (INCa-DGOS-INSERM_12563), Agence Nationale de la Recherche (FR) Institut Convergence PLASCA, (17-CONV-0002), Agence Nationale de la Recherche (FR) LabEx DEvweCAN (ANR-10-LABX-0061), Agence Nationale de la Recherche (FR) RHU4 DEPGYN (ANR-18-RHUS-0009) Institut National Du Cancer (FR) INTERSARC, Fondation ARC pour la Recherche sur le Cancer Unicancer, Ligue Contre le Cancer (FR) Unicancer, Ligue Contre le Cancer Comité de l'Ain Canopee, and by EUROPEAN COMMISSION (EU) EURACAN (EC 739521). The funders had no role in study design, data collection and analysis, decision to publish, or preparation of the manuscript.

**Competing interests:** No competing interests.

# Abstract

## Background

Since 2010, nationwide networks of reference centers for sarcomas (RREPS/NETSARC/RESOS) collected and prospectively reviewed all cases of sarcomas and connective tumors of intermediate malignancy (TIM) in France.

## Methods

The nationwide incidence of sarcoma or TIM (2013–2016) was measured using the 2013 WHO classification and confirmed by a second independent review by expert pathologists. Simple clinical characteristics, yearly variations and correlation of incidence with published clinical trials are presented and analyzed.

## Results

Over 150 different histological subtypes are reported from the 25172 patients with sarcomas (n = 18712, 74,3%) or TIM (n = 6460, 25.7%), with n = 5838, n = 6153, n = 6654, and n = 6527 yearly cases from 2013 to 2016. Over these 4 years, the yearly incidence of sarcomas and TIM was therefore 70.7 and 24.4 respectively, with a combined incidence of $95.1/10^6$/year, higher than previously reported. GIST, liposarcoma, leiomyosarcomas, undifferentiated sarcomas represented 13%, 13%, 11% and 11% of tumors. Only GIST, as a single entity had a yearly incidence above $10/10^6$/year. There were respectively 30, 64 and 66 different histological subtypes of sarcomas or TIM with an incidence ranging from 10 to $1/10^6$, $1–0.1/10^6$, or $< 0.1/10^6$/year respectively. The 2 latter incidence groups represented 21% of the patients with 130 histotypes. Published phase III and phase II clinical trials ($p<10^{-6}$) are significantly higher with sarcomas subtypes with an incidence above $1/10^6$ per.

## Conclusions

This nationwide registry of sarcoma patients, with exhaustive histology review by sarcoma experts, shows that the incidence of sarcoma and TIM is higher than reported, and that tumors with a very low incidence ($1<10^6$/year) are less likely to be included in clinical trials.

## Introduction

Sarcomas are a group of rare malignant diseases of the connective tissues, with heterogeneous clinical presentations and natural histories. The incidence of sarcoma was reported 15 years ago to be close to $2/10^5$/year, but more recently, the global incidence was reported to be higher, ranging from 3 to 7 /$10^5$/year [1–16]. The incidence of sarcoma varies across countries and according to the date of the study [2–16] and is therefore not precisely known. Similarly, the incidence of each individual histological subtypes is unclear and sarcomas are misdiagnosed in up to 30% of cases [1, 5, 6, 11, 17]. These patients with misdiagnosed sarcomas may not be treated according to clinical practice guidelines [1–21]. Clinical practice guidelines recommend that the diagnosis of sarcoma should be confirmed by an expert pathologist and that the management of sarcoma patients should be performed by a dedicated multidisciplinary team,

including expert pathologists and surgeons, treating a minimal number of patients per year [5–7]. Central pathology review of sarcoma cases was shown to be cost effective, reducing both morbidity, mortality and cost of management [22, 23].

Since 2010, the French National Cancer Institute (INCa) funded a pathology network (RRePS) and a clinical network (NETSARC) for sarcoma, to improve the quality of management of sarcoma patients. Subsequently, a dedicated bone sarcoma network named RESOS was created. Initially, the network composed of 23 expert reference centers for pathology (RRePS) was in charge of the histological review of each suspected case of sarcoma nationwide. Since 2019, all three networks have merged in a single network (acronym NETSARC+). The shared online database (see rreps.org and netsarc.org) gathers all cases of sarcoma reviewed by a multidisciplinary tumor board. It collects data related to diagnostic, therapeutic management, relapse and survival.

Since the January 1$^{st}$ 2010, this database prospectively has included over 60000 patients with sarcoma and connective tissue tumor of intermediate malignancy (TIM), as defined according to the WHO 2013 classification [1]. Since 2013, the overall accrual in the database reached a plateau, providing a near exhaustive collection of cases in this country.

The global nationwide incidence of sarcoma has seldom been reported. Taking advantage of the organized reference center networks and expert pathology reviewing, we report here the incidence of the different histological subtypes of sarcomas and TIM from the NETSARC + database from 2013 to 2016.

## Patient, material and methods

### The NETSARC+ network and the referral of the pathology samples to the network of experts

The RRePS (an acronym standing for Reseau de Relecture en Pathologie des Sarcomes, i.e. Network of Sarcoma Pathology Reviewing, gathering 23 centers), NetSarc (Network for Sarcoma, in charge of clinical management, gathering 26 centers), and ResOs (Network for Bone Sarcomas, dedicated to bone sarcoma pathology and clinical network) networks were merged into NETSARC+ (Network of Sarcomas Reference Centers) in 2019.

The organization of these networks has been previously reported [24, 25]: each RRePS and NetSarc centers hold a multidisciplinary tumor board gathering sarcoma specialized pathologist(s) (S1 Table), radiologist(s), surgeon(s), radiation oncologist(s), medical oncologist(s), molecular biologist(s), orthopedist(s) and pediatrician(s).

The missions given by the French National Cancer institute (INCa) funding the networks were, among others, to review all pathology samples and to collect a set of anonymized data to monitor patient outcome. Since 2010, it is mandatory for the primary pathologist to refer all suspected cases of sarcomas or TIM to one of these reference centers.

### The RRePS/NetSarc database

These databases have been approved by the French Ethic Committee and Agency in charge of non-interventional trials: Comité consultatif sur le traitement de l'information en matière de recherche dans le domaine de la Santé (CCTIRS: number of approval 09.594) and Commission Nationale Informatique et Liberté (CNIL: number of approval 909510). THe consent was obtained orally.

All sarcoma/TIM or suspected sarcoma/TIM patient cases discussed during the multidisciplinary tumor board (MDTB) of all 26 centers were recorded in the electronic online database, by a dedicated team of Clinical research assistant (CRAs), supervised by the Coordinating

centers (Unicancer Comprehensive Cancer Centers: Centre Leon Bérard, Gustave Roussy, Institut Bergonié, as well as CHU Tours & CHU Nantes). Patients files may be presented at different stages of care process, before any diagnostic procedure, before initial biopsy, before primary surgery, after primary surgery, at relapse, and/or in case of a possible inclusion in a clinical trial as previously described [24, 25]. All patients had the option to opt out the initiave, or could orally consent to the pathology review of their biopsy, the registry in the national network database according to the national laws at that time, as well as benefit from the recommendations of the French National Cancer Institute.

The databases were therefore not generated from clinical trial data, and we built to monitor and improve the management of patients with sarcomas in France. According to the national legislation, the activity of the three networks did not have to be reviewed by the national ethics committees, compared to clinical studies (the Comité de Protection des Personnes). However, each local hospitals (i.e. the reference centers) used their internal procedure (internal institutional review board) to approve participation to this work. This approval was mandatory for the participation of the given center to any of the 3 networks.

Oral consent is documented through a standard information sheet given to all patients in each institution in the welcome leaflet. Patients are able to opt out, and this information is taken into account during multidisciplinary tumor board discussions of the cases.

Patients and treatment data were prospectively collected and regularly updated by the dedicated study coordinators. The cases were obtained directly from the pathologist laboratories, or referred to the expert pathology laboratory for diagnosis confirmation for a fraction of the patients who were first seen by the clinical MDT. This double source of entry contributed to improve the exhaustiveness of the collection of cases. Of note, the database includes on purpose a limited set of data, describing for example patients and tumor characteristics, surgery, relapse and survival [24, 25], centers performing the first resection, as well as potential secondary surgery types and sites, the final quality of resection.

The RRePS/NetSarc database may therefore give a nearly exhaustive representation of sarcoma cases to assess sarcoma incidence and prevalence in France.

The database is not systematically updated for follow-up by the CRA, as for clinical trial processes, but all baseline and first therapeutic information are completed until the end of the first line treatment. This includes all pathology reviews, which are therefore as presented, the final diagnoses, with a median follow-up of the series of 17 months in the recent publications of the same dataset (25). Importantly, since 2019, the data (survival and all treatments) from the nationwide database of the social security system (SNDS, https://www.snds.gouv.fr/SNDS/Accueil) is used to update the latest survival information of these patients as part of the Health DataHub Deepsarc project (https://www.health-data-hub.fr/outil-de-visualisation), now ensuring an exhaustive follow-up information.

Of note, the diagnosis of sarcomas of TIM (e.g. lipoma, carcinoma, lymphoma...) was not confirmed by the NetSarc MDTB for about 24% of the patients (not shown).

All data presented here were extracted from the common online database available online for a period of 4 years between 2013 and 2016. These 4 years were selected since: 1) the yearly incidence of sarcoma and TIM started to plateau since 2013, and 2) data monitoring and implementation is still ongoing since 2017.

## Presentation of the data

The 2013 WHO classification of sarcomas and connective tissue tumors was used from January 1$^{st}$ 2013 to describe the histological subtypes in the database, taking advantage of the contribution of the 2013 version (since April 2012) by one of the senior authors (JMC) of the

current article, [1]. Monthly physical meetings of the pathologist network to review complex cases have facilitated the homogeneity of data collection within this group. Both sarcomas and TIM were included in the database. Tumors of intermediate malignancy designates connective tissue tumors with the capacity to invade the surrounding tissues, with very rare metastases (1). These include for instance, aggressive fibromatosis, dermatofibrosarcoma protuberans, atypical lipomatous tumors . . .

The number of patients for each individual histological subtype of sarcoma or TIM per year, from 2013 to 2016, is therefore presented in the tables. To facilitate the comparison with other databases using previous classifications, the incidence of tumor groups (e.g. "uterine sarcomas") are also presented in the tables. Each individual histotype, (e.g. WDLPS) and groups of histotypes are presented when clinically relevant (e.g. "WD and DDLPS", or "liposarcoma", "leiomyosarcomas"). Conversely, when a grouping did not exert relevance in clinical routine (e.g. "fibroblastic and myofibroblastic tumors" in Table 1), no new entity was described.

To estimate the incidence of these tumors, we used the official number of French citizens from 2013 to 2016, which were respectively 65.56, 66.13, 66.42 and 66.60 million inhabitants.

## Matching histotypes with published clinical trials

For each individual histotype, we searched Pubmed to identify published dedicated clinical trials. The name of the histological entity (e.g. angiosarcoma, pleomorphic liposarcoma. . .) was filtered for clinical trial, adding « phase III », « randomized phase II », or « phase II ». Pubmed was interrogated between January 15th 2020 and January 30th 2020. For the presentation of these data, all sarcoma histotypes or groups of histotypes, were ranked according to the order of decreasing incidence. When at least one phase II (light blue), one randomized phase II (dark blue), or one phase III (green) was published in the literature this was indicated in the line of the histotype using this colour code. It was also also used for the 3 columns of sarcoma/TIM with decreasing incidences to facilitate the visibility of the correlation between incidence of sarcoma and availability of clinical trials on the figure.

## Statistical analyses

The number of patients per year with the different histotypes is presented in tables. To analyze the variation of incidence over the 4 years, a Poisson Regression was used. Six histotypes with a significant variation in the period of observation are graphically detailed by overlaying the observed incidence, a linear regression over time and a spline. The comparison of the frequency of published clinical trials per histological subtypes or groups of subtypes was performed using the chi square or Fisher's exact test with a threshold p value of $p < 0.05$. All statistical tests were two-sided. All statistical analyses were performed using SPSS (v 23.0) (IBM, Paris, France).

## Results

### Incidence of sarcoma and TIM in NetSarc

Tables 1–3 present the incidence of the individual histological subtypes of soft tissue sarcomas/TIM (Table 1), visceral sarcomas/TIM (Tables 1 and 3 for uterine sarcoma), bone sarcomas/TIM (Tables 2 and 3) included in the database (gathering the RRePS, NetSarc, and ResOs) from 2013 to 2016, a period where the data are expected to be close to exhaustive nationwide.

From 2013 to 2016, a total of 25172 patients were included in the database, with n = 5838, n = 6153, n = 6654, and n = 6527 of new patients each year. Of note, respectively N = 4435 N = 4977 and N = 5550 patients were included in each year from 2010 to 2012 (not shown).

**Table 1. Incidence of visceral and soft tissue sarcoma and tumors of intermediate malignancy over the 4 years of the study.**

| | 2013 | 2014 | 2015 | 2016 | Total | Incidence /10^6/year |
|---|---|---|---|---|---|---|
| **Adipocytic tumours** | **744** | **821** | **817** | **865** | **3247** | **12,299** |
| Atypical lipomatous tumour / | | | | | | |
| well-differentiated liposarcoma | 289 | 304 | 314 | 357 | 1266 | 4,795 |
| Liposarcoma–dedifferentiated | 304 | 344 | 341 | 356 | 1345 | 5,095 |
| **Myxoid Round Cell LPS** | **99** | **106** | **108** | **96** | **409** | **1,549** |
| Liposarcoma—myxoid | 81 | 90 | 95 | 89 | 355 | 1,345 |
| Liposarcoma—round cell | 18 | 16 | 13 | 7 | 54 | 0,205 |
| Liposarcoma—pleomorphic | 31 | 41 | 36 | 31 | 139 | 0,527 |
| Lipomatous spindle cell/pleomorphic | 0 | 1 | 0 | 0 | 1 | 0,004 |
| Liposarcoma NOS | 21 | 25 | 17 | 22 | 85 | 0,322 |
| Liposarcoma—mixed type | 0 | 0 | 1 | 1 | 2 | 0,008 |
| **Fibroblastic & myofibroblastic tumours** | **1041** | **1047** | **1115** | **1147** | **4349** | **16,473** |
| Desmoid fibromatosis | 307 | 295 | 357 | 381 | 1340 | 5,072 |
| Lipofibromatosis | 3 | 0 | 5 | 0 | 8 | 0,030 |
| Giant cell Fibroblastoma | 2 | 4 | 4 | 1 | 11 | 0,042 |
| Dermatofibrosarcoma Protuberans | 261 | 270 | 258 | 251 | 1040 | 3,939 |
| **Solitary fibrous tumour (all)** | **210** | **222** | **242** | **252** | **925** | **3,504** |
| • Solitary fibrous tumor | 166 | 178 | 193 | 214 | 751 | 2,845 |
| • High risk SFT | 44 | 43 | 49 | 38 | 174 | 0,659 |
| Inflammatory myofibroblastic Tum. | 32 | 39 | 33 | 41 | 145 | 0,549 |
| Low grade Myofibroblastic Sarc. | 3 | 5 | 3 | 2 | 13 | 0,049 |
| Myxoinflammatory Fibroblastic Sarc. | 6 | 6 | 6 | 5 | 23 | 0,087 |
| Infantile fibrosarcoma | 3 | 2 | 1 | 4 | 10 | 0,038 |
| Adult fibrosarcoma | 11 | 4 | 9 | 4 | 28 | 0,106 |
| Myxofibrosarcoma | 162 | 160 | 152 | 156 | 630 | 2,386 |
| Low grade fibromyxoid sarcoma | 33 | 30 | 35 | 38 | 136 | 0,515 |
| Sclerosing epithelioid fibrosarcoma | 8 | 11 | 10 | 12 | 41 | 0,155 |
| **So-called fibrohistiocytic tumours** | **29** | **16** | **37** | **24** | **106** | **0,402** |
| Intermediate fibrohistiocytic tumors (NOS) | 0 | 0 | 2 | 3 | 5 | 0,019 |
| Malignant tenosynovial giant cell tum. | 1 | 0 | 0 | 1 | 2 | 0,008 |
| Plexiform fibrohistiocytic tumors | 7 | 7 | 9 | 6 | 29 | 0,110 |
| Giant cell tumour of soft tissue | 21 | 9 | 26 | 14 | 70 | 0,265 |
| **Vascular tumours** | **398** | **377** | **381** | **364** | **1520** | **5,758** |
| Retiform hemangio-endothelioma | 1 | 3 | 3 | 2 | 9 | 0,034 |
| Papillary intralymphatic | | | | | | |
| angioendothelioma | 0 | 0 | 0 | 1 | 1 | 0,004 |
| Composite hemangioendothelioma | 1 | 1 | 1 | 0 | 3 | 0,011 |
| Kaposi sarcoma | 191 | 165 | 162 | 145 | 663 | 2,511 |
| Kaposiform hemangioendothelioma | 1 | 1 | 1 | 1 | 4 | 0,015 |
| Pseudomyogenic hemangioendothelioma | 1 | 3 | 0 | 2 | 6 | 0,023 |
| Epithelioid hemangioEndothelioma | 27 | 20 | 30 | 23 | 100 | 0,379 |
| Angiosarcoma | 176 | 183 | 182 | 187 | 728 | 2,758 |
| Intermediate vascular tumours (NOS) | 0 | 1 | 2 | 3 | 6 | 0,023 |
| **Pericytic (perivascular) tumours** | **4** | **4** | **1** | **1** | **10** | **0,038** |
| Malignant glomus tumour | | | | | | |
| **Smooth muscle (SM) tumours** | **646** | **698** | **669** | **666** | **2679** | **10,148** |

*(Continued)*

**Table 1.** (Continued)

| | | | | | Total | Incidence |
|---|---|---|---|---|---|---|
| SM tumor of undetermined malignancy | 20 | 47 | 23 | 32 | 122 | 0,462 |
| Metastatic leiomyoma | 0 | 0 | 0 | 2 | 2 | 0,008 |
| **Leiomyosarcoma (All)** | **626** | **651** | **646** | **632** | **2555** | **9,679** |
| Leiomyosarcoma (NOS) | 247 | 263 | 287 | 297 | 1094 | 4,144 |
| Leiomyosarcoma -differentiated | 245 | 243 | 240 | 217 | 945 | 3,580 |
| Leiomyosarcoma–poorly differentiated | 134 | 145 | 119 | 118 | 516 | 1,955 |
| **Skeletal muscle sarcoma (RMS)** | **145** | **157** | **173** | **133** | **608** | **2,303** |
| **Embryonal RMS** | **50** | **45** | **60** | **34** | **189** | **0,716** |
| • Embryonal RMS sarcoma—botryoid type | 8 | 6 | 6 | 3 | 23 | 0,087 |
| • Embryonal rhabdomyosarcoma usual type | 35 | 31 | 47 | 24 | 137 | 0,519 |
| • Embryonal rhabdomyosarcoma spindle cell | 7 | 8 | 7 | 7 | 29 | 0,110 |
| Alveolar RMS | 27 | 36 | 35 | 25 | 123 | 0,466 |
| Pleomorphic RMS | 28 | 38 | 42 | 36 | 144 | 0,545 |
| Sclerosing RMS | 2 | 3 | 3 | 3 | 11 | 0,042 |
| Spindle cell RMS | 13 | 8 | 9 | 9 | 39 | 0,148 |
| Adult spindle cell RMS | 0 | 0 | 1 | 4 | 5 | 0,019 |
| RMS NOS | 21 | 25 | 23 | 19 | 88 | 0,333 |
| Ectomesenchymoma: Mal. mesenchymoma | 4 | 2 | 0 | 3 | 9 | 0,034 |
| **Gastrointestinal stromal tumors (GIST).** | **736** | **792** | **913** | **831** | **3272** | **12,394** |
| **Chondro-osseous tumours** | | | | | | |
| Extraskeletal osteosarcoma | **25** | **25** | **32** | **14** | **96** | **0,364** |
| **Peripheral nerve sheath tumours** | **75** | **68** | **69** | **74** | **286** | **1,083** |
| **MPNST (all)** | **72** | **64** | **62** | **66** | **264** | **1.000** |
| MPNST—epithelioid type | 0 | 2 | 1 | 3 | 6 | 0,023 |
| MPNST—usual type | 36 | 7 | 14 | 28 | 85 | 0,322 |
| Malignant peripheral nerve sheath tumour | 36 | 55 | 47 | 35 | 173 | 0,655 |
| Malignant Triton tumour | 0 | 2 | 3 | 5 | 10 | 0,038 |
| Malignant granular cell Tumour | 3 | 2 | 4 | 0 | 9 | 0,034 |
| Malignant perineurioma | 0 | 0 | 0 | 3 | 3 | 0,011 |
| | **2013** | **2014** | **2015** | **2016** | **Total** | **Incidence** |
| | | | | | | **/10e6/year** |
| **Tumours of uncertain differentiation** | | | | | | |
| Atypical fibroxanthoma | 114 | 107 | 89 | 119 | 429 | 1,625 |
| Angiomatoid fibrous histiocytoma | 9 | 15 | 10 | 9 | 43 | 0,163 |
| Ossifying fibromyxoid Tumour | 7 | 7 | 5 | 13 | 32 | 0,121 |
| **Myoepithelioma, myoepithelial carcinoma,** | | | | | | |
| **& mixed tumour** | **31** | **26** | **18** | **18** | **93** | **0,353** |
| • Myoepithelioma | 30 | 26 | 15 | 14 | 85 | 0,322 |
| • Malignant myoepithelial Tumour | 0 | 0 | 1 | 1 | 2 | 0,008 |
| • Mixed tumour | 1 | 0 | 2 | 3 | 6 | 0,023 |
| Haemosiderotic fibrolipomatous tumour | 0 | 2 | 0 | 7 | 9 | 0,034 |
| Phosphaturic mesenchymal tumour | 0 | 1 | 2 | 2 | 5 | 0,019 |
| NTRK-rearranged spindle cell neoplasm (emerging) Not reported in NETSARC (so far) | | | | | | |
| **Synovial sarcoma** | **103** | **101** | **133** | **105** | **442** | **1,674** |
| • Synovial sarcoma–NOS | 23 | 18 | 29 | 21 | 91 | 0,345 |
| • Synovial sarcoma—biphasic | 11 | 19 | 23 | 17 | 70 | 0,265 |
| • Synovial sarcoma–monophasic | 60 | 57 | 67 | 60 | 244 | 0,924 |
| • Synovial sarcoma—poorly Differentiated | 9 | 7 | 14 | 7 | 37 | 0,140 |

*(Continued)*

**Table 1.** (Continued)

| | | | | | | |
|---|---|---|---|---|---|---|
| **Epithelioid sarcoma (all)** | **29** | **30** | **28** | **33** | **120** | **0,455** |
| • Epithelioid sarcoma | 23 | 28 | 25 | 22 | 98 | 0,371 |
| • Undifferentiated epithelioid sarcoma | 6 | 2 | 3 | 11 | 22 | 0,083 |
| Alveolar soft part sarcoma | 10 | 7 | 8 | 6 | 31 | 0,117 |
| Clear cell sarcoma of soft tissue | 13 | 16 | 26 | 16 | 71 | 0,269 |
| Extraskeletal myxoid chondrosarcoma | 15 | 12 | 20 | 11 | 58 | 0,220 |
| Desmoplastic small round cell tumour | 14 | 9 | 12 | 17 | 52 | 0,197 |
| Extrarenal rhabdoid tumour | 6 | 13 | 16 | 16 | 51 | 0,193 |
| SMARCA4-deficient thoracic sarcoma | 0 | 0 | 6 | 9 | 15 | 0,057 |
| **PEComa, including angiomyolipoma** | **13** | **27** | **15** | **29** | **86** | **0,326** |
| • PECOMA—NOS | 13 | 25 | 11 | 18 | 67 | 0,254 |
| • Malignant PECOMA | 0 | 2 | 4 | 13 | 19 | 0,072 |
| Intimal sarcoma | 14 | 12 | 11 | 9 | 46 | 0,174 |
| **Undifferentiated sarcoma (all)** | **566** | **627** | **784** | **740** | **2717** | **10,292** |
| • Undifferentiated pleomorphic sarcoma | 290 | 367 | 470 | 429 | 1556 | 5,894 |
| • Undifferentiated sarcoma | 110 | 87 | 154 | 79 | 430 | 1,629 |
| • Undifferentiated sarcoma -NOS | 125 | 130 | 111 | 57 | 423 | 1,602 |
| • Undifferentiated spindle cell sarcoma | 41 | 43 | 49 | 175 | 308 | 1,167 |
| Low grade sinonasal sarcoma | 2 | 0 | 0 | 3 | 5 | 0,019 |
| Melanotic neuroectodermal tumour infancy | 0 | 0 | 0 | 1 | 1 | 0,004 |
| **Phyllode sarcoma** | **32** | **25** | **46** | **35** | **138** | **0,523** |
| **Sarcomas or TIM NOS** | | | | | | |
| Sarcoma NOS | 166 | 197 | 259 | 189 | 809 | 3,064 |
| Tumors of intermediate malignancy (NOS) | 7 | 15 | 13 | 17 | 52 | 0,197 |

In red, groups of tumors (e.g. adipocytic tumors) according to the WHO classification, in bold, subgroups of tumors considered relevant (e.g. "all leiomyosarcomas", "All MPNST"). Note that in few of these patients with sarcomas and TIM mostly of soft tissue /visceral origin, the primary site was indicated as being from the bone. They are described specifically in Table 3. Phyllodes tumors refer to all malignant phyllodes.

The NetSarc database contains over 150 individual histological subtypes (i.e. a single histological entity such as monophasic synovial sarcoma, low grade surface osteosarcoma, atypical lipomatous tumor) or groups of sarcomas (e.g. liposarcoma, leiomyosarcoma, osteosarcoma, where the grouping of individual histological entity when clinically relevant) (Tables 1–). The grouping are described in S2 Table. Twelve additional histological subtypes of bone sarcomas (leiomyosarcomas, synovial etc) were also distinguished in this work and described in Table 3. These histotypes usually arising from soft tissue are also included in Table 1. Finally, Table 3 also presents the incidence of sarcomas diagnosed in patients with reported genetic predispositions, such as Li Fraumeni syndrome.

The official numbers of the French population are 65.56, 66.13, 66.42 and 66.60 million inhabitants in respectively 2013, 2014, 2015, 2016. The estimated incidences of sarcomas and tumors of intermediate malignancy from 2013 to 2016 were 89.05, 93.04, 100.18, and 98.00 per million inhabitants respectively. Over these 4 years, the estimated yearly incidence of sarcomas and TIM was therefore $95,1/10^6$/year. There were 18712 (74%) patients with sarcomas (incidence $70.7/10^e$/year) and 6460 (26%, $24.4/10^6$/year) patients with TIM. The observed overall incidence of sarcoma and TIMs is therefore higher that previously reported [1–15].

**Table 2. Incidence of bone sarcoma over the 4 years of the study.**

| | 2013 | 2014 | 2015 | 2016 | Total | Incidence |
|---|---|---|---|---|---|---|
| | | | | | | /10⁶/year |
| **Undifferentiated small round cell sarcomas** | | | | | | |
| **(SRCS) of bone and soft tissue** | | | | | | |
| Ewing sarcoma | 151 | 163 | 153 | 147 | 614 | 2,326 |
| SRCS with EWSR1-non-ETS fusions | 6 | 6 | 8 | 36 | 56 | 0,212 |
| CIC-rearranged sarcoma | 1 | 3 | 3 | 4 | 11 | 0,042 |
| BCOR-rearranged Sarcoma | 2 | 0 | 2 | 3 | 7 | 0,027 |
| **Bone tumours** | | | | | | |
| **Chondrogenic tumours** | | | | | | |
| Chondroblastoma | 11 | 16 | 9 | 16 | 52 | 0,197 |
| Chondromyxoid fibroma | 4 | 7 | 12 | 3 | 26 | 0,098 |
| **Chondrosarcoma (all)** | **227** | **211** | **244** | **285** | **967** | **3,663** |
| Central atypical cartilaginous | | | | | | |
| tumour/chondrosarcoma, gd 1 | 2 | 10 | 19 | 45 | 76 | 0,288 |
| Central chondroS grades 2 and 3 | 22 | 33 | 35 | 27 | 117 | 0,443 |
| Chondrosarcoma NOS | 164 | 125 | 143 | 140 | 572 | 2,167 |
| Peripheral chondrosarcoma | 5 | 6 | 8 | 20 | 39 | 0,148 |
| Periosteal chondrosarcoma | 8 | 4 | 6 | 7 | 25 | 0,095 |
| Clear cell chondrosarcoma | 4 | 3 | 2 | 5 | 14 | 0,053 |
| Mesenchymal chondrosarcoma | 3 | 7 | 11 | 10 | 31 | 0,117 |
| Dedifferentiated chondrosarcoma | 19 | 23 | 20 | 31 | 93 | 0,352 |
| **Osteogenic tumors** | | | | | | |
| Osteoblastoma | 5 | 9 | 8 | 10 | 32 | 0,121 |
| **Osteosarcoma (all)** | **330** | **362** | **370** | **376** | **1438** | **7,122** |
| Low grade central osteosarcoma | 4 | 4 | 4 | 7 | 19 | 0,072 |
| Low-grade central osteosarcoma | 2 | 1 | 1 | 3 | 7 | 0,027 |
| Dediff. low grade central osteosarcoma | 2 | 3 | 3 | 4 | 12 | 0,045 |
| Osteosarcoma | 154 | 169 | 170 | 168 | 661 | 2,504 |
| Osteosarcoma NOS | 39 | 57 | 62 | 72 | 230 | 1,106 |
| Conventional osteosarcoma | 111 | 105 | 105 | 96 | 417 | 1,580 |
| Osteoblastoma-like osteosarcoma | 1 | 0 | 0 | 1 | 2 | 0,008 |
| Telangiectasic osteosarcoma | 2 | 7 | 2 | 5 | 16 | 0,061 |
| Small cell osteosarcoma | 1 | 0 | 1 | 2 | 4 | 0,015 |
| **Parosteal osteosarcoma** | **7** | **7** | **16** | **10** | **40** | **0,152** |
| • Parosteal osteosarcoma | 6 | 4 | 10 | 7 | 27 | 0,102 |
| • Dedifferentiated parosteal osteoSarc | 1 | 3 | 6 | 3 | 13 | 0,049 |
| Periosteal osteosarcoma | 1 | 4 | 0 | 0 | 5 | 0,019 |
| High-grade surface osteosarcoma | 6 | 5 | 6 | 8 | 25 | 0,095 |
| **Fibrogenic tumors** | | | | | | |
| Desmoplastic fibroma of bone | 0 | 0 | 2 | 4 | 6 | 0,023 |
| Fibrosarcoma of the bone | 0 | 1 | 3 | 0 | 4 | 0,015 |
| **Vascular tumor of bone** | | | | | | |
| Epithelioid haemangioendothelioma | 1 | 2 | 5 | 0 | 8 | 0,030 |
| Angiosarcoma of bone | 7 | 6 | 7 | 9 | 29 | 0,110 |
| **Osteoclastic giant-cell rich** | | | | | | |
| Aneurysmal bone cyst | 14 | 9 | 22 | 8 | 53 | 0,201 |
| Giant cell tumour of bone | 76 | 88 | 87 | 67 | 318 | 1,204 |

*(Continued)*

**Table 2.** (Continued)

| | 2013 | 2014 | 2015 | 2016 | Total | Incidence |
|---|---|---|---|---|---|---|
| Malignant/dedifferentiated GCTB | 0 | 0 | 2 | 4 | 6 | 0,023 |
| **Notochordal tumors** | | | | | | |
| • Conventional chordoma | 35 | 33 | 42 | 54 | 164 | 0,621 |
| • Dedifferentiated chordoma | 0 | 1 | 1 | 0 | 2 | 0,008 |
| **Adamantinoma** | 8 | 1 | 2 | 8 | 19 | 0,072 |
| **Langerhans cell histiocytosis** | 4 | 8 | 4 | 4 | 20 | 0,076 |

In red, groups of tumors (e.g. chondrogenic tumors) according to the WHO classification, in bold, subgroups of tumors considered relevant (e.g. all osteosarcoma).

## Over 100-fold difference in incidence in different sarcoma histotypes

To complete Table 1 data, S1 Fig presents the individual histotypes and relevant groups of histotypes (e.g. liposarcoma, leiomyosarcoma, uterine sarcomas) by increasing incidence. GIST, liposarcoma, leiomyosarcomas, undifferentiated sarcomas represent 13%, 13%, 11% and 11% of all sarcomas (47% all together). Only gastrointestinal stromal tumors, if considered as a single entity, exceeded a yearly incidence above $10/10^6$/ year (S1 Fig). The other histological types of sarcomas with a yearly incidence above $10/10^6$/year are 1) all liposarcomas, 2) all smooth muscle tumors, 3) all undifferentiated sarcomas, and 4) all fibroblastic or myofibroblastic tumors lumped together. This latter group is not clinically homogenous and usually not considered as a specific entity in clinical trials or retrospective studies.

Fig 1 presents the list of sarcomas or TIM with a decreasing incidence ranging from 10 to 1/$10^6$/year, 1–0.1/$10^6$ per year, or $<0.1/10^6$/year. The 3 groups include respectively 30, 64 and 66 different histological subtypes or groups of histological subtypes. The groups with an incidence from 1–0.1/$10^6$ per year, or $<0.1/10^6$/year included respectively 4766 (19%) and 568 (2%) of the 25172 patients.

The mean age, sex ratio and sites for the different histotypes are presented in Table 4. It shows the large clinical heterogeneity of these tumors with a mean age ranging from 5 years (infantile fibrosarcoma) to 78 (atypical fibroxanthoma), and a sex ratio from 0 (for sexual organs) to 153 for adenosarcoma.

## Variable incidence of sarcoma histotypes over the 2013–2016 period

We investigated then the variability of the yearly incidence of these different tumors in the database. The analysis of variance of the observed incidence indicated a significant interaction between time and histology. S2 Fig presents the six histological subtypes with the mst significant variation between 2013 and 2016. Adenosarcoma, desmoid tumors, malignant pecoma, UPS, endometrial stromal sarcoma—high-grade increased over the 4-year period, while myoepithelioma showed a decrease of incidence (S2 Fig). The significance of these variations remains unclear and needs further investigation using comparable registries with a centralized review.

## Incidence of individual histotypes and published clinical trials

Table 1 and S1 Fig gives a graphic presentation in decreasing order of the incidence of the different histotypes and groups of histotypes. These were matched with published clinical trial data collected from Pubmed on a given histological group (e.g. liposarcoma) or specific histotype (e.g. pleomorphic liposarcoma). Phase III studies, randomized phase II studies and non-randomized phase II studies are indicated in green, dark blue and light blue respectively

**Table 3. Incidence of uterine and rare bone sarcoma over the 4 years of the study.**

| | 2013 | 2014 | 2015 | 2016 | Total | Incidence /10e6/year |
|---|---|---|---|---|---|---|
| **Uterine sarcoma** | **242** | **311** | **285** | **300** | **1138** | **4,311** |
| **Endometrial stromal sarcoma, low grade (all)** | **57** | **64** | **55** | **62** | **238** | **0,902** |
| • Endometrial stromal nodule | 2 | 1 | 5 | 8 | 16 | 0,061 |
| • Endometrial stromal sarcoma | 0 | 3 | 2 | 5 | 10 | 0,038 |
| • Endometrial stromal sarcoma-low grade | 55 | 60 | 48 | 49 | 212 | 0,803 |
| Endometrial stromal sarcoma—high-grade | 1 | 5 | 13 | 22 | 41 | 0,155 |
| Adenosarcoma | 28 | 35 | 42 | 51 | 156 | 0,591 |
| Undifferentiated uterine sarcoma | 37 | 49 | 38 | 17 | 141 | 0,534 |
| Uterine tumour resembling ovarian sex cord | 5 | 1 | 4 | 7 | 17 | 0,064 |
| **Uterine leiomyosarcoma** | | | | | | |
| (extracted from the LMS group above) | **114** | **157** | **133** | **141** | **545** | **2,064** |
| **Rare bone sarcomas** | | | | | | |
| (extracted from the histological groups in Table 1) | | | | | | |
| **All undifferentiated sarcoma of bone** | **36** | **31** | **38** | **47** | **152** | **0,576** |
| • Undifferentiated pleomorphic sarcoma | 16 | 20 | 12 | 21 | 69 | 0,261 |
| • Undifferentiated sarcoma | 16 | 11 | 21 | 8 | 56 | 0,212 |
| • Undifferentiated spindle cell sarcoma | 4 | 0 | 5 | 17 | 26 | 0,098 |
| • Undifferentiated epithelioid sarcoma | 0 | 0 | 0 | 1 | 1 | 0,004 |
| Leiomyosarcoma of bone | 11 | 15 | 5 | 9 | 40 | 0,152 |
| Synovial sarcoma of bone | 4 | 2 | 2 | 1 | 9 | 0,034 |
| Rhabdomyosarcoma of bone | 2 | 2 | 1 | 4 | 9 | 0,034 |
| BCOR Sarcoma of bone | 1 | 0 | 2 | 3 | 6 | 0,023 |
| Myoepithelioma of bone | 1 | 1 | 1 | 1 | 4 | 0,015 |
| Liposarcoma of bone | 0 | 2 | 0 | 2 | 4 | 0,015 |
| Other histological subtypes of bone sarcomas | 33 | 42 | 46 | 50 | 171 | 0,648 |
| **Genetic predisposition of soft tissue and bone or HIV** | | | | | | |
| Enchondromatosis | 5 | 2 | 6 | 4 | 17 | 0,064 |
| Li Fraumeni syndrome | 3 | 3 | 4 | 4 | 14 | 0,053 |
| Retinoblastoma | 1 | 0 | 3 | 1 | 5 | 0,019 |
| Multiple osteochondroma | 2 | 2 | 11 | 5 | 20 | 0,076 |
| Neurofibromatosis | 28 | 28 | 24 | 25 | 105 | 0,398 |
| Rothmund-Thomson | 0 | 1 | 0 | 0 | 1 | 0,004 |
| HIV | 4 | 12 | 10 | 6 | 32 | 0,121 |
| Other immunosuppression | 3 | 13 | 4 | 5 | 25 | 0,095 |

In red, groups of tumors (e.g. low grade endometrial stromal sarcoma). UTROSC were identified by the network experts and included in the database: though borderline for this group of diseases, we chose to insert them in this list.

showing a variable access to clinical trial according to the incidence of the histotype. An histological subtype is considered "covered" by a trial only if the trial design contains a specific arm (phase II) or a specific strata (phase III) for a given histotype.

As expected, phase III trials are made available mostly for histotypes or groups of histotypes with an incidence $>1/10^6$ per year (Fig 1). 14 of 35 (40%) histotypes and groups of histotypes with an incidence $>1/10^6$ had a dedicated phase III study vs 6 of 130 (4.6%) histotypes for sarcomas with an incidence $<1/10^6$ ($p<10^{-6}$). 20100 (79,7%) patients of the database had a histotype for which no phase III trial had been reported. Twenty-one of 35 (60%) histotypes with an

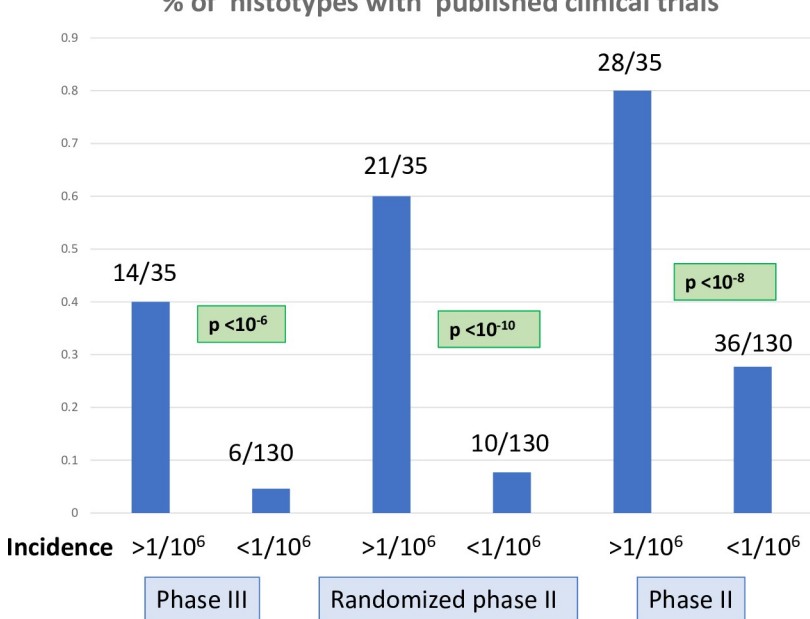

**Fig 1. Published clinical trials in sarcoma and TIM histotypes.** The histograms present the percentage of sarcoma histotypes and groups of histotypes with published clinical trials in Pubmed according to the incidence of the histotypes ($>1/10^6$/year vs $<1/10^6$/year). Numbers of histotypes with published phase III clinical trials (left), randomized phase II trials (center), and phase II trials (right) are indicated, with chi square p value for the comparison between the 2 incidence groups ($>1/10^6$/year vs $<1/10^6$/year). Histotypes were considered individually (e.g. monophasic synovial sarcoma) or globally (e.g. all synovial sarcoma).

incidence $>1/10^6$ had a dedicated randomized phase II study vs 10 of 130 (7.7%) histotypes for sarcomas with a incidence $<1/10^6$ ($p<10^{-10}$). 13154 (52.1%) patients of the database had a specific histotype for which no randomized phase II trial had been reported. Twenty-eight of 35 (80%) histotypes with an incidence $>1/10^6$ had a dedicated phase II study vs 36 of 130 (27.9%) histotypes for sarcomas with an incidence $<1/10^6$ ($p<10^{-8}$). 6516 (25.8%) patients of the database had a specific histotype for which no phase II trial had been reported.

## Discussion

The objective of this work was to measure the incidence of individual histological subtypes of sarcomas and TIM according to the 2013 WHO classification. These cases were collected from the NETSARC+ database, combining the previous RRePS, ResOs and NetSarc databases. This work, supported by the French NCI, allowed to measure the incidence of sarcomas and TIM in a nationwide level. The mandatory central pathology review, in place since 2010, has facilitated the constitution of a close to exhaustive nationwide collection of patients with sarcoma and TIM. Since 2013, the number of patients included in the database per year is relatively stable suggesting that the database may indeed be close to exhaustiveness. We stopped the description in 2016, since the period spanning from 2017 to 2019 is still being monitored and updated by the CRAs of NETSARC+.

The first important observation is that the incidence of these tumors is higher than previously reported in each of these 4 years [1–15]. Recently published data from countries in 4 different continents reported an overall incidence ranging from 3 to $7.7/10^6$/year. The results of these studies are heterogeneous in terms of proportion distribution of the histotypes, ranging from 4 to 20% for undifferentiated sarcomas for instance. These observations suggest that

**Table 4. Clinical characteristics of individual sarcoma histotypes.**

| Histotypes | Mean Age | F/M Ratio | Sites (%)* | | | | | | | |
|---|---|---|---|---|---|---|---|---|---|---|
| | | | GI | Gyn | H&N | I. trnk | L. limb | Trnk w | U. limb | Others |
| Adamantinoma | 29,1 | 1,71 | 0,0 | 0,0 | 0,0 | 0,0 | 100,0 | 0,0 | 0,0 | 0,0 |
| Adenosarcoma | 61,3 | 51,00 | 1,9 | 89,7 | 0,0 | 3,8 | 0,6 | 0,6 | 0,0 | 3,2 |
| Adult fibrosarcoma | 69,5 | 1,15 | 0,0 | 7,1 | 7,1 | 10,7 | 17,9 | 28,6 | 17,9 | 10,7 |
| Adult spindle cell rhabdomyosarcoma | 52,4 | 0,25 | 0,0 | 0,0 | 20,0 | 0,0 | 20,0 | 40,0 | 20,0 | 0,0 |
| Alveolar rhabdomyosarcoma | 22,7 | 1,05 | 0,0 | 0,8 | 40,7 | 16,3 | 13,8 | 9,8 | 13,8 | 4,9 |
| Alveolar soft part sarcoma | 30,5 | 1,21 | 0,0 | 0,0 | 9,7 | 6,5 | 51,6 | 22,6 | 6,5 | 3,2 |
| Aneurysmal bone cyst | 30,2 | 1,12 | 0,0 | 0,0 | 3,8 | 1,9 | 39,6 | 32,1 | 20,8 | 1,9 |
| Angiomatoid fibrous histiocytoma | 27,1 | 0,87 | 0,0 | 0,0 | 7,0 | 11,6 | 37,2 | 18,6 | 25,6 | 0,0 |
| Angiosarcoma | 67,2 | 1,57 | 1,2 | 0,5 | 14,1 | 7,0 | 9,9 | 21,4 | 3,3 | 42,4 |
| Atypical cartilaginous Tumour/ChondroS G1 | 45,0 | 1,38 | 0,0 | 0,0 | 3,9 | 0,0 | 36,8 | 14,5 | 43,4 | 1,3 |
| Atypical fibroxanthoma | 78,7 | 0,20 | 0,2 | 0,0 | 12,8 | 0,0 | 1,6 | 82,1 | 1,6 | 1,6 |
| Atypical lipomatous tumor/WDLPS | 64,2 | 0,84 | 0,9 | 0,1 | 2,7 | 29,8 | 44,3 | 14,9 | 5,8 | 1,4 |
| BCOR sarcoma | 18,9 | 0,40 | 0,0 | 0,0 | 0,0 | 28,6 | 28,6 | 42,9 | 0,0 | 0,0 |
| Central chondrosarcoma | 56,4 | 0,98 | 0,0 | 0,0 | 7,7 | 13,7 | 35,0 | 17,1 | 26,5 | 0,0 |
| Chondroblastoma | 24,2 | 0,44 | 0,0 | 0,0 | 5,8 | 0,0 | 63,5 | 17,3 | 13,5 | 0,0 |
| Chondromyxoid fibroma | 34,1 | 0,73 | 0,0 | 0,0 | 3,8 | 0,0 | 42,3 | 30,8 | 23,1 | 0,0 |
| Chondrosarcoma NOS | 54,1 | 0,91 | 0,3 | 0,3 | 13,5 | 20,6 | 25,5 | 21,3 | 17,0 | 1,4 |
| Chordoma | 61,7 | 0,71 | 0,0 | 0,0 | 13,4 | 0,6 | 0,0 | 85,4 | 0,0 | 0,6 |
| CIC-DUX sarcoma | 24,1 | 0,38 | 9,1 | 0,0 | 9,1 | 0,0 | 45,5 | 27,3 | 0,0 | 9,1 |
| Clear cell chondrosarcoma | 42,5 | 0,27 | 0,0 | 0,0 | 0,0 | 7,1 | 64,3 | 7,1 | 21,4 | 0,0 |
| Clear cell sarcoma | 42,3 | 0,87 | 12,7 | 0,0 | 5,6 | 2,8 | 46,5 | 14,1 | 14,1 | 4,2 |
| Composite hemangioendothelioma | 33,3 | 0,50 | 0,0 | 0,0 | 0,0 | 0,0 | 33,3 | 66,7 | 0,0 | 0,0 |
| Conventional osteosarcoma | 32,6 | 0,80 | 0,0 | 0,0 | 12,5 | 3,1 | 60,2 | 14,4 | 9,6 | 0,2 |
| Dedifferentiated chondrosarcoma | 64,0 | 0,94 | 0,0 | 0,0 | 1,1 | 11,8 | 48,4 | 30,1 | 7,5 | 1,1 |
| Dedifferentiated chordoma | 53,0 | NA | 0,0 | 0,0 | 0,0 | 0,0 | 0,0 | 100,0 | 0,0 | 0,0 |
| Dedifferentiated low-grade central osteo | 36,2 | 1,40 | 0,0 | 0,0 | 0,0 | 0,0 | 83,3 | 8,3 | 8,3 | 0,0 |
| Dedifferentiated parosteal osteosarcoma | 43,8 | 2,25 | 0,0 | 0,0 | 7,7 | 0,0 | 76,9 | 7,7 | 7,7 | 0,0 |
| Dermatofibrosarcoma protuberans | 45,0 | 1,02 | 0,0 | 0,1 | 3,0 | 0,5 | 19,7 | 61,3 | 13,8 | 1,5 |
| Desmoid-type fibromatosis | 43,8 | 2,17 | 15,9 | 0,1 | 3,3 | 7,7 | 7,4 | 59,0 | 6,3 | 0,4 |
| Desmoplastic fibroma of bone | 27,5 | 1,00 | 0,0 | 0,0 | 0,0 | 0,0 | 66,7 | 16,7 | 16,7 | 0,0 |
| Desmoplastic round cell tumour | 24,3 | 0,30 | 3,8 | 0,0 | 0,0 | 84,6 | 0,0 | 0,0 | 0,0 | 11,5 |
| Embryonal rhabdomyosarcoma -botryoid type | 10,7 | 2,29 | 0,0 | 43,5 | 21,7 | 0,0 | 0,0 | 0,0 | 0,0 | 34,8 |
| Embryonal rhabdomyosarcoma—NOS | 20,3 | 0,71 | 0,0 | 16,7 | 41,7 | 8,3 | 0,0 | 8,3 | 0,0 | 25,0 |
| Embryonal rhabdomyosarcoma—spindle cell | 19,0 | 0,45 | 0,0 | 6,9 | 31,0 | 37,9 | 0,0 | 6,9 | 3,4 | 13,8 |
| Embryonal rhabdomyosarcoma—usual type | 14,4 | 0,57 | 0,9 | 4,4 | 35,4 | 38,9 | 3,5 | 3,5 | 0,9 | 12,4 |
| Endometrial stromal nodule | 50,7 | NA | 0,0 | 81,3 | 0,0 | 0,0 | 0,0 | 18,8 | 0,0 | 0,0 |
| Endometrial stromal sarcoma NOS | 56,2 | NA | 0,0 | 80,0 | 0,0 | 20,0 | 0,0 | 0,0 | 0,0 | 0,0 |
| Endometrial stromal sarcoma—high-grade | 60,0 | NA | 0,0 | 95,1 | 0,0 | 4,9 | 0,0 | 0,0 | 0,0 | 0,0 |
| Endometrial stromal sarcoma—low-grade | 53,0 | 211,00 | 0,5 | 88,7 | 0,0 | 9,4 | 0,0 | 0,5 | 0,0 | 0,9 |
| Epithelioid hemangioendothelioma | 52,1 | 1,56 | 2,0 | 0,0 | 9,0 | 11,0 | 14,0 | 14,0 | 5,0 | 45,0 |
| Epithelioid sarcoma | 40,1 | 0,85 | 1,0 | 6,1 | 2,0 | 12,2 | 22,4 | 19,4 | 29,6 | 7,1 |
| Ewing sarcoma | 26,0 | 0,68 | 1,0 | 0,5 | 5,7 | 16,6 | 28,2 | 33,6 | 7,8 | 6,7 |
| Extraskeletal myxoid chondrosarcoma | 58,2 | 0,81 | 0,0 | 0,0 | 0,0 | 1,7 | 58,6 | 27,6 | 8,6 | 3,4 |
| Extraskeletal osteosarcoma | 63,1 | 0,78 | 0,0 | 1,0 | 1,0 | 1,0 | 44,8 | 24,0 | 19,8 | 8,3 |
| Fibro-osseous tumour of bone NOS | 36,0 | NA | 0,0 | 0,0 | 0,0 | 0,0 | 0,0 | 100,0 | 0,0 | 0,0 |

(*Continued*)

**Table 4.** (Continued)

| Histotypes | | | Sites (%)* | | | | | | | |
| --- | --- | --- | --- | --- | --- | --- | --- | --- | --- | --- |
| | Mean | F/M | | | | | | | | |
| | Age | Ratio | GI | Gyn | H&N | I. trnk | L. limb | Trnk w | U. limb | Others |
| Fibrosarcomatous dermatofibrosarcoma prot. | 45,6 | 0,65 | 0,0 | 0,0 | 3,4 | 1,7 | 18,8 | 65,0 | 10,3 | 0,9 |
| Gastrointestinal stromal tumour (GIST), | 65,4 | 0,94 | 94,8 | 0,1 | 0,0 | 4,8 | 0,0 | 0,2 | 0,0 | 0,1 |
| Giant cell fibroblastoma | 22,0 | 0,38 | 0,0 | 0,0 | 0,0 | 0,0 | 54,5 | 45,5 | 0,0 | 0,0 |
| Giant cell tumour of bone | 37,8 | 1,13 | 0,0 | 0,0 | 0,9 | 0,0 | 58,3 | 14,4 | 25,1 | 1,3 |
| Giant cell tumour of soft tissues | 47,5 | 1,41 | 0,0 | 0,0 | 7,1 | 1,4 | 47,1 | 8,6 | 35,7 | 0,0 |
| Hemosiderotic fibrolipomatous tumour | 45,4 | 3,50 | 0,0 | 0,0 | 0,0 | 0,0 | 100,0 | 0,0 | 0,0 | 0,0 |
| High risk solitary fibrous tumour | 64,4 | 0,85 | 2,9 | 0,6 | 8,0 | 31,0 | 6,3 | 13,8 | 1,7 | 35,6 |
| High-grade surface osteosarcoma | 44,6 | 0,92 | 0,0 | 0,0 | 24,0 | 12,0 | 44,0 | 20,0 | 0,0 | 0,0 |
| Infantile fibrosarcoma | 5,9 | 2,33 | 10,0 | 0,0 | 20,0 | 10,0 | 20,0 | 20,0 | 0,0 | 20,0 |
| Inflammatory myofibroblastic tumour | 39,3 | 1,10 | 11,0 | 4,1 | 11,7 | 54,5 | 6,2 | 6,9 | 5,5 | 0,0 |
| Intermediate fibrohistiocytic tumours NOS | 41,0 | 0,25 | 0,0 | 0,0 | 0,0 | 0,0 | 60,0 | 0,0 | 40,0 | 0,0 |
| Intermediate vascular tumours NOS | 64,7 | 5,00 | 0,0 | 0,0 | 16,7 | 0,0 | 0,0 | 83,3 | 0,0 | 0,0 |
| Intimal sarcoma | 58,9 | 0,92 | 0,0 | 0,0 | 0,0 | 39,1 | 2,2 | 0,0 | 0,0 | 58,7 |
| Kaposi sarcoma | 65,8 | 0,22 | 1,1 | 0,0 | 3,2 | 1,1 | 65,8 | 11,5 | 13,4 | 4,1 |
| Kaposiform hemangioendothelioma | 6 | 3,00 | 0,0 | 0,0 | 0,0 | 0,0 | 50,0 | 50,0 | 0,0 | 0,0 |
| Langerhans cell histiocytosis | 29,5 | 4,00 | 0,0 | 0,0 | 10,0 | 5,0 | 5,0 | 75,0 | 5,0 | 0,0 |
| Leiomyosarcoma | 63,5 | 2,18 | 4,8 | 35,3 | 7,1 | 18,7 | 15,1 | 8,0 | 4,9 | 5,9 |
| Leiomyosarcoma—differentiated | 63,1 | 1,23 | 4,9 | 18,2 | 6,1 | 24,2 | 18,4 | 11,5 | 9,7 | 6,9 |
| Leiomyosarcoma—poorly-differentiated | 70,3 | 0,73 | 3,7 | 8,9 | 29,7 | 9,7 | 19,4 | 15,3 | 7,6 | 5,8 |
| Lipofibromatosis | 10,3 | 1,00 | 12,5 | 0,0 | 12,5 | 0,0 | 12,5 | 50,0 | 12,5 | 0,0 |
| Lipomatous spindle cell/pleomorphic tum | 33,0 | NA | 0,0 | 0,0 | 0,0 | 0,0 | 100,0 | 0,0 | 0,0 | 0,0 |
| Liposarcoma—dedifferentiated | 67,9 | 0,60 | 2,0 | 0,3 | 1,6 | 69,5 | 11,4 | 9,5 | 2,2 | 3,5 |
| Liposarcoma—mixed type | 61,0 | 1,00 | 0,0 | 0,0 | 0,0 | 50,0 | 50,0 | 0,0 | 0,0 | 0,0 |
| Liposarcoma—myxoid | 47,8 | 0,81 | 0,3 | 0,3 | 0,0 | 4,8 | 77,5 | 14,4 | 2,3 | 0,6 |
| Liposarcoma—NOS | 64,2 | 0,57 | 1,2 | 1,2 | 2,4 | 38,8 | 31,8 | 11,8 | 8,2 | 4,7 |
| Liposarcoma—pleomorphic | 63,1 | 0,78 | 0,7 | 0,7 | 3,6 | 15,1 | 38,1 | 22,3 | 15,8 | 3,6 |
| Liposarcoma—round cell | 49,2 | 0,59 | 0,0 | 0,0 | 0,0 | 14,8 | 63,0 | 14,8 | 7,4 | 0,0 |
| Low grade fibromyxoid sarcoma | 42,5 | 0,97 | 0,0 | 0,0 | 9,6 | 5,9 | 34,6 | 36,8 | 11,0 | 2,2 |
| Low grade myofibroblastic sarcoma | 40,5 | 1,17 | 7,7 | 0,0 | 46,2 | 0,0 | 23,1 | 23,1 | 0,0 | 0,0 |
| Low grade sinonasal sarcoma | 37,6 | 1,50 | 0,0 | 20,0 | 80,0 | 0,0 | 0,0 | 0,0 | 0,0 | 0,0 |
| Low-grade central osteosarcoma | 33,6 | 2,50 | 0,0 | 0,0 | 0,0 | 14,3 | 71,4 | 0,0 | 14,3 | 0,0 |
| Malignant glomus tumour | 54,2 | 0,67 | 20,0 | 10,0 | 10,0 | 10,0 | 30,0 | 0,0 | 20,0 | 0,0 |
| Malignant granular cell tumour | 46,1 | 1,25 | 0,0 | 0,0 | 0,0 | 0,0 | 0,0 | 55,6 | 44,4 | 0,0 |
| Malignant mesenchymoma | 61,1 | 1,25 | 0,0 | 11,1 | 0,0 | 11,1 | 33,3 | 0,0 | 33,3 | 11,1 |
| Malignant mixed tumor | 67,5 | NA | 25,0 | 75,0 | 0,0 | 0,0 | 0,0 | 0,0 | 0,0 | 0,0 |
| Malignant myoepithelial tumour | 49,5 | 1,00 | 0,0 | 0,0 | 0,0 | 0,0 | 0,0 | 50,0 | 50,0 | 0,0 |
| Malignant PECOMA | 60,1 | 1,71 | 15,8 | 21,1 | 0,0 | 26,3 | 10,5 | 10,5 | 0,0 | 15,8 |
| Malignant perineurioma | 46,3 | 2,00 | 0,0 | 0,0 | 0,0 | 0,0 | 33,3 | 33,3 | 0,0 | 33,3 |
| Malignant peripheral nerve sheath tumour | 46,4 | 0,86 | 0,6 | 0,0 | 11,6 | 17,3 | 24,9 | 31,2 | 9,8 | 4,6 |
| Malignant rhabdoid tumour | 24,3 | 0,89 | 2,8 | 8,3 | 13,9 | 16,7 | 5,6 | 16,7 | 0,0 | 36,1 |
| Malignant tenosynovial giant cell tumour | 68,5 | 0,00 | 0,0 | 0,0 | 0,0 | 50,0 | 0,0 | 50,0 | 0,0 | 0,0 |
| Malignant Triton tumour | 34,7 | 1,00 | 0,0 | 0,0 | 30,0 | 40,0 | 0,0 | 30,0 | 0,0 | 0,0 |
| Malignant/dedifferentiated giant cell tumor of the bone | 40,0 | 1,00 | 0,0 | 0,0 | 0,0 | 0,0 | 80,0 | 0,0 | 20,0 | 0,0 |
| Melanotic neuroectodermal tumour of infa | 38,0 | NA | 0,0 | 0,0 | 0,0 | 100,0 | 0,0 | 0,0 | 0,0 | 0,0 |
| Mesenchymal chondrosarcoma | 34,9 | 0,72 | 0,0 | 0,0 | 29,0 | 12,9 | 29,0 | 25,8 | 0,0 | 3,2 |

(*Continued*)

**Table 4.** (Continued)

| Histotypes | Mean Age | F/M Ratio | Sites (%)* | | | | | | | |
|---|---|---|---|---|---|---|---|---|---|---|
| | | | GI | Gyn | H&N | I. trnk | L. limb | Trnk w | U. limb | Others |
| Metastatic leiomyoma | 39,0 | NA | 0,0 | 0,0 | 0,0 | 50,0 | 0,0 | 50,0 | 0,0 | 0,0 |
| Mixed tumour | 66,0 | NA | 0,0 | 0,0 | 0,0 | 50,0 | 50,0 | 0,0 | 0,0 | 0,0 |
| MPNST—epithelioid type | 43,2 | 2,00 | 0,0 | 0,0 | 0,0 | 16,7 | 16,7 | 50,0 | 16,7 | 0,0 |
| MPNST—usual type | 45,4 | 0,77 | 1,2 | 1,2 | 14,1 | 12,9 | 27,1 | 27,1 | 11,8 | 4,7 |
| Myoepithelioma | 50,6 | 0,89 | 0,0 | 0,0 | 3,5 | 3,5 | 37,6 | 27,1 | 25,9 | 2,4 |
| Myxofibrosarcoma | 68,9 | 0,70 | 0,2 | 0,0 | 2,7 | 2,1 | 46,0 | 18,6 | 28,4 | 2,1 |
| Myxoinflammatory fibroblastic sarcoma | 54,3 | 0,53 | 0,0 | 0,0 | 0,0 | 0,0 | 39,1 | 0,0 | 60,9 | 0,0 |
| Osseous tumour rich in giant cell NOS | 40,5 | 1,00 | 0,0 | 0,0 | 0,0 | 0,0 | 50,0 | 50,0 | 0,0 | 0,0 |
| Ossifying fibromyxoid tumour | 49,8 | 1,13 | 0,0 | 0,0 | 9,4 | 6,3 | 12,5 | 37,5 | 34,4 | 0,0 |
| Osteoblastoma | 26,5 | 0,48 | 0,0 | 0,0 | 3,2 | 0,0 | 19,4 | 58,1 | 19,4 | 0,0 |
| Osteoblastoma-like osteosarcoma | 29,0 | NA | 0,0 | 0,0 | 0,0 | 0,0 | 100,0 | 0,0 | 0,0 | 0,0 |
| Osteogenic tumor of uncertain prognosis | 22,0 | NA | 0,0 | 0,0 | 0,0 | 0,0 | 0,0 | 100,0 | 0,0 | 0,0 |
| Osteosarcoma NOS | 38,3 | 0,64 | 0,0 | 0,0 | 15,7 | 2,6 | 51,3 | 19,1 | 9,1 | 2,2 |
| Papillary intralymphatic angioendothelioma | 13,0 | NA | 0,0 | 0,0 | 0,0 | 0,0 | 0,0 | 100,0 | 0,0 | 0,0 |
| Parosteal osteosarcoma | 33,6 | 2,86 | 0,0 | 0,0 | 0,0 | 3,7 | 85,2 | 0,0 | 11,1 | 0,0 |
| PECOMA—NOS | 55,7 | 3,79 | 11,9 | 25,4 | 3,0 | 41,8 | 6,0 | 4,5 | 1,5 | 6,0 |
| Periosteal chondrosarcoma | 41,3 | 1,08 | 4,0 | 0,0 | 0,0 | 8,0 | 32,0 | 24,0 | 32,0 | 0,0 |
| Periosteal osteosarcoma | 19,8 | 4,00 | 0,0 | 0,0 | 0,0 | 20,0 | 80,0 | 0,0 | 0,0 | 0,0 |
| Peripheral chondrosarcoma | 37,3 | 0,63 | 0,0 | 0,0 | 0,0 | 10,3 | 33,3 | 33,3 | 23,1 | 0,0 |
| Phosphaturic mesenchymal tumour | 55,8 | 0,67 | 0,0 | 0,0 | 0,0 | 0,0 | 40,0 | 60,0 | 0,0 | 0,0 |
| Phyllodes sarcoma | 51,3 | 137,00 | 0,0 | 0,0 | 0,0 | 1,4 | 0,0 | 13,0 | 0,0 | 85,5 |
| Pleomorphic rhabdomyosarcoma | 67,0 | 0,58 | 1,4 | 7,6 | 5,6 | 11,8 | 36,1 | 19,4 | 11,1 | 6,9 |
| Plexiform fibrohistiocytic tumour | 23,1 | 1,23 | 0,0 | 0,0 | 6,9 | 3,4 | 34,5 | 24,1 | 31,0 | 0,0 |
| Pseudomyogenic hemangioendothelioma | 37,3 | 1,00 | 0,0 | 0,0 | 0,0 | 0,0 | 50,0 | 50,0 | 0,0 | 0,0 |
| Retiform hemangioendothelioma | 40,2 | 0,80 | 0,0 | 0,0 | 0,0 | 0,0 | 22,2 | 44,4 | 33,3 | 0,0 |
| Rhabdomyosarcoma—NOS | 41,7 | 0,80 | 1,1 | 13,6 | 17,0 | 23,9 | 11,4 | 4,5 | 5,7 | 22,7 |
| Sclerosing epithelioid fibrosarcoma | 55,8 | 1,05 | 0,0 | 0,0 | 9,8 | 17,1 | 14,6 | 41,5 | 9,8 | 7,3 |
| Sclerosing rhabdomyosarcoma | 45,2 | 0,38 | 0,0 | 0,0 | 9,1 | 0,0 | 72,7 | 0,0 | 9,1 | 9,1 |
| Small cell osteosarcoma | 21,8 | 0,33 | 0,0 | 0,0 | 0,0 | 25,0 | 25,0 | 0,0 | 25,0 | 25,0 |
| SMARCA4-deficient thoracic sarcoma | 48,3 | 0,25 | 13,3 | 0,0 | 0,0 | 40,0 | 0,0 | 0,0 | 0,0 | 46,7 |
| Smooth muscle tumour of undetermined mal | 50,6 | 4,30 | 6,6 | 59,8 | 0,8 | 15,6 | 5,7 | 7,4 | 4,1 | 0,0 |
| Solitary fibrous tumour | 58,0 | 1,25 | 1,2 | 1,1 | 15,4 | 21,0 | 13,3 | 18,6 | 4,3 | 25,0 |
| Spindle cell rhabdomyosarcoma | 38,8 | 0,63 | 5,1 | 2,6 | 17,9 | 20,5 | 28,2 | 10,3 | 12,8 | 2,6 |
| Suspicion of giant cell tumour of bone | 56,3 | 0,00 | 0,0 | 0,0 | 0,0 | 0,0 | 33,3 | 0,0 | 66,7 | 0,0 |
| Sarcoma NOS | 59,2 | 1,03 | 4,7 | 7,7 | 11,0 | 13,3 | 21,8 | 18,0 | 11,4 | 12,1 |
| Synovial sarcoma—biphasic | 41,2 | 0,84 | 1,4 | 0,0 | 5,7 | 5,7 | 52,9 | 17,1 | 10,0 | 7,1 |
| Synovial sarcoma—monophasic | 42,8 | 1,18 | 1,2 | 0,0 | 4,9 | 9,4 | 42,6 | 13,1 | 15,6 | 13,1 |
| Synovial sarcoma NOS | 45,6 | 1,22 | 0,0 | 1,1 | 5,5 | 8,8 | 34,1 | 13,2 | 16,5 | 20,9 |
| Synovial sarcoma—poorly differentiated | 45,2 | 0,85 | 0,0 | 0,0 | 5,4 | 10,8 | 29,7 | 18,9 | 8,1 | 27,0 |
| Telangiectasic osteosarcoma | 25,1 | 0,60 | 0,0 | 0,0 | 6,3 | 0,0 | 75,0 | 0,0 | 18,8 | 0,0 |
| Tumour of intermediate malignancy NOS | 47 | 1,14 | 5,8 | 3,8 | 13,5 | 13,5 | 21,2 | 19,2 | 21,2 | 1,9 |
| Undifferentiated epithelioid sarcoma | 70 | 0,47 | 0,0 | 0,0 | 13,6 | 18,2 | 22,7 | 13,6 | 27,3 | 4,5 |
| Undifferentiated pleomorphic sarcoma | 69,2 | 0,79 | 1,7 | 0,8 | 15,6 | 7,6 | 35,0 | 19,6 | 14,0 | 5,7 |
| Undifferentiated round cell sarcoma | 41,2 | 1,33 | 1,8 | 5,4 | 10,7 | 16,1 | 23,2 | 23,2 | 3,6 | 16,1 |
| Undifferentiated sarcoma | 66,7 | 0,89 | 1,4 | 3,3 | 15,8 | 13,0 | 29,5 | 21,4 | 8,8 | 6,7 |

*(Continued)*

**Table 4.** (*Continued*)

| Histotypes | | | | | | Sites (%)* | | | | |
|---|---|---|---|---|---|---|---|---|---|---|
| | Mean | F/M | | | | | | | | |
| | Age | Ratio | GI | Gyn | H&N | I. trnk | L. limb | Trnk w | U. limb | Others |
| Undifferentiated sarcoma—NOS | 62,5 | 0,86 | 1,2 | 4,5 | 23,2 | 14,4 | 18,9 | 18,2 | 6,1 | 13,5 |
| Undifferentiated spindle cell sarcoma | 64,1 | 0,79 | 1,6 | 2,3 | 19,8 | 11,7 | 23,7 | 19,8 | 9,7 | 11,4 |
| Undifferentiated uterine sarcoma | 63,8 | NA | 0,0 | 96,1 | 0,0 | 2,9 | 0,0 | 1,0 | 0,0 | 0,0 |
| Uterine tumour resembling ovarian sex co | 48,1 | NA | 5,9 | 94,1 | 0,0 | 0,0 | 0,0 | 0,0 | 0,0 | 0,0 |

*: Sites: GI: gastrointestinal; Gyn: Gynaecological sites; H&N: head and neck; I. trnk: Internal trunk; L. Limb: lower limb; Trnk w: trunk wall; U. limb: upper limb. Sarcoma histotypes are presented by alphabetical order. Age, sex ratio, and groups of sites are presented for each histotypes.

mandatory central pathology review, implemented during this period, enabled to collect exhaustive data to assess the incidence of almost all subtypes. In addition, it relied on expert reviewing and therefore a more accurate diagnosis.

Importantly, there are no national registries for any cancers, including sarcomas in France and there are several limitations for using regional registries for the model of rare cancers. These limitations are 1) the already mentioned importance of centralized expert pathology review in sarcoma, which corrects the first diagnosis in >20% of the cases [5–7, 16], and 2) the lack of exhaustiveness of these registries for sarcomas; an ongoing collaborative work between NetSarc and the network of Regional registries (unpublished) indicates that the incidence of sarcoma and TIM is less than 70% of that observed in the present report. An increased collaboration is strongly needed between these structures.

Mandatory central pathology review is essential for these rare cancers and it can be achieved through several pathways: 1) each pathology review report indicates the mandatory review process, the sites as well as the contact information of the expert pathologists; all French pathologists are addressed this report multiple times per year; 2) when missed, patients presented in clinical MDT without pathology review are immediately referred to the closest expert pathology center; 3) the patients themselves exerted on multiple occasions their request for a pathology review, asking for a wide dissemination on the internet of the existence of networks of excellence and the mandatory pathology and clinical review process.

While the incidence of sarcoma and TIM observed here is higher than previously reported [1–15], it is important to point out that comparison with historical series has important limitations: some individual entities described here may have been described only very recently (e.g. GIST before 1999, solitary fibrous tumors, etc...). The numbers presented are those measured with a more recent classification which has dramatically evolved in recent decades.

The present work also confirms that sarcoma and TIM are a highly fragmented group of diseases. Over 150 different histotypes or groups of histotypes relevant for clinical practice are listed. The precise definition of this number was somehow challenging, and in part subjective for the selection of a relevant group of histotypes treated homogenously in routine in clinical trials (e.g. "liposarcoma"). It must be noted also that the actual number of different disease entities may actually be larger. GIST, for instance, gathers tumors with completely different genetic somatic alterations (of *KIT*, *PDGFRA*, *SDH*, *NF1*, *BRAF*, *NTRK3*) each with different natural histories.

The individual incidence of different sarcoma histotypes ranged from $10/10^6$ to less than $0.01/10^6$, i.e. a >1000-fold difference in incidence. Altogether sarcomas are considered as rare

cancer, but the majority of individual subtypes are actually extremely rare. This is of course challenging for the understanding of the natural history of each individual subtypes, for the development of clinical trials, new treatments as well as defining standards of care.

Sarcoma and TIM are also a highly heterogeneous group of disease for clinical presentations as shown by the diversity of sex ratio and mean age of diagnosis. Each of these entities should therefore benefit from a specific research program to describe their natural history as well as the impact of current treatment on their disease course. This requires a coordinated effort, worldwide, to achieve this goal given the rarity of certain histotypes. This is currently being conducted by intergroup studies, and international networks such as WSN or more recently EURACAN [26–28]. This work also confirms the importance of national registries to investigate these rare subtypes.

An intriguing observation is the variation of incidence for these tumors over time, and statistically significant for several histotypes. There is however no clear explanation for this observation. Given the stable position of pathology experts involved in the network, this may not likely result from the variable pathology review. Epidemiological studies in other countries may be useful to confirm these variations, which may guide research on etiology of these rare sarcomas and TIM.

Another observation, expected by clinicians, is the link between the incidence and the availability of published prospective clinical research work to guide the management of individual subtypes. For decades, the medical treatment of sarcomas used a one-size-fits-all approach for all histotypes from phase II to III clinical trials, in particular in adults. For 15 years, histotype specific randomized phase II, III and phase II studies were implemented, starting with GIST. This is more the exception of the rule though. The majority of histotypes described in this work, especially those with an incidence under $1/10^6$/year have not had a dedicated phase II, randomized phase II or phase III clinical trial to adapt or guide clinical practice guidelines. This is graphically obvious, even though the mode of presentation of the figure amplified this phenomenon, since we presented both individual subtypes (e.g. monophasic synovial sarcoma, with no dedicated clinical trials to our knowledge) and pooled histotypes (e.g. all synovial sarcomas, where clinical trials were listed). It must be noted, however, that in the group with an incidence $<0.1/10^6$/year, some histotypes were included in dedicated clinical trials. Due to the rarity of these tumors, clinical trials are expected to only be feasible at a global worldwide level, and randomization will not be feasible.

Overall, this calls for a revision of the criteria to define standard treatment for such rare tumors where phase III are hardly or not feasible [26–29]. Health authorities and reimbursement bodies should adapt their decisions on approval and reimbursement on the feasible level of evidence which could be reached for tumors with and incidence $<1/10^6$ per year in order not to discriminate against patients with rare cancers. It is important to remember that altogether patients with rare cancers represent 22% of all patients with cancers, and about 30% of the mortality due to cancer [29].

This study has many limitations. We cannot exclude that patients may not reach our network despite the administrative incentive. This is true in particular for bone sarcoma and TIM (e.g. chondroblastomas, osteoblastoma, aneurysmatic bone cyst, etc.) which were collected more recently.

The work must also adapt to the rapidly evolving classification of sarcomas, including new molecular sub-classifications, which are not described here (for instance GIST, the novel NTRK sarcoma subgroup, BCOR, CIC-DUX4 sarcomas). To further explore the exhaustivity of the NETSARC+ database, an ongoing project connects it to the social security data base (SNDS) (the Health DataHub Deepsarc project), the single payer in France covering all citizens, across all diseases. This should enable to enhance the accuracy of these numbers.

In conclusion, this nationwide registry reports a higher than previously reported incidence of sarcoma and TIM at a nationwide level, over a 4-year period, with a central sarcoma pathologist expertise review. Our data provide a benchmark to be compared with other worldwide registries and confirm the limitations of clinical research in sarcomas with an incidence inferior to $1/10^6$ per year. The observation of variable incidence for specific histological subtypes is intriguing and requires further investigation by using data from other countries. Geographical research on the distribution of these cases over the national territory are currently ongoing.

## Supporting information

**S1 Table. Contributing pathologists of all centers RRePS & RESOS.** This table includes per alphabetical order of institutions, the different pathologists contributing to this work. Note that a single reference center from RREPS or now NETSARC+ may include more than one institution.
(DOCX)

**S2 Table. Codes of the histotypes according to WHO 2013.**
(DOCX)

**S3 Table. Grouping of histotypes presented in the tables.**
(DOCX)

**S1 Fig. Published clinical trials in sarcoma and TIM histotypes.** Tabular presentation of different sarcoma histotypes and groups of histotypes by decreasing order together with the documented published clinical trials in Pubmed. If phase III clinical trials are published, the box is highlighted in light green, if randomized phase II trials are published the box is highlighted in dark blue, if uncontrolled phase II trials are published the box is highlighted in light blue. Histotypes were considered individually (e.g. monophasic synovial sarcoma) or globally (e.g. all synovial sarcoma).
(TIF)

**S2 Fig. Variable incidence of sarcoma subtypes from 2013 to 2016.** Presentation of the yearly variation of six different histotypes with significantly variable incidence in the period of observation.
(TIF)

## Acknowledgments

The authors thank Stephanie Cox and Muriel Rogasik for their review of the manuscript.

## Author Contributions

**Conceptualization:** Gonzague de Pinieux, Marie Karanian, Axel Le Cesne, Nicolas Penel, Francoise Ducimetiere, Francois Gouin, Jean-Michel Coindre, Jean-Yves Blay.

**Data curation:** Marie Karanian, Francois Le Loarer, Sophie Le Guellec, Sylvie Chabaud, Philippe Terrier, Corinne Bouvier, Maxime Batistella, Agnès Neuville, Yves-Marie Robin, Jean-Francois Emile, Anne Moreau, Frederique Larousserie, Agnes Leroux, Nathalie Stock, Marick Lae, Francoise Collin, Nicolas Weinbreck, Sebastien Aubert, Florence Mishellany, Celine Charon-Barra, Sabrina Croce, Laurent Doucet, Isabelle Quintin-Rouet, Marie-Christine Chateau, Celine Bazille, Isabelle Valo, Bruno Chetaille, Nicolas Ortonne, Anne Brouchet, Philippe Rochaix, Anne Demuret, Jean-Pierre Ghnassia, Lenaig Mescam, Nicolas Macagno, Isabelle Birtwisle-Peyrottes, Christophe Delfour, Emilie Angot, Isabelle

Pommepuy, Dominique Ranchere, Claire Chemin-Airiau, Myriam Jean-Denis, Yohan Fayet, Maud Toulmonde, Antoine Italiano, Nicolas Penel, Francoise Ducimetiere, Francois Gouin, Jean-Michel Coindre, Jean-Yves Blay.

**Formal analysis:** Axel Le Cesne, Nicolas Penel, Francois Gouin, Jean-Michel Coindre, Jean-Yves Blay.

**Funding acquisition:** Gonzague de Pinieux, Maud Toulmonde, Antoine Italiano, Axel Le Cesne, Nicolas Penel, Francoise Ducimetiere, Francois Gouin, Jean-Michel Coindre, Jean-Yves Blay.

**Investigation:** Gonzague de Pinieux, Marie Karanian, Francois Le Loarer, Sophie Le Guellec, Sylvie Chabaud, Philippe Terrier, Corinne Bouvier, Maxime Batistella, Agnès Neuville, Yves-Marie Robin, Jean-Francois Emile, Anne Moreau, Frederique Larousserie, Agnes Leroux, Nathalie Stock, Marick Lae, Francoise Collin, Nicolas Weinbreck, Sebastien Aubert, Florence Mishellany, Celine Charon-Barra, Sabrina Croce, Laurent Doucet, Isabelle Quintin-Rouet, Marie-Christine Chateau, Celine Bazille, Isabelle Valo, Bruno Chetaille, Nicolas Ortonne, Anne Brouchet, Philippe Rochaix, Anne Demuret, Jean-Pierre Ghnassia, Lenaig Mescam, Nicolas Macagno, Isabelle Birtwisle-Peyrottes, Christophe Delfour, Emilie Angot, Isabelle Pommepuy, Dominique Ranchere, Yohan Fayet, Maud Toulmonde, Antoine Italiano, Francoise Ducimetiere, Francois Gouin, Jean-Michel Coindre, Jean-Yves Blay.

**Methodology:** Gonzague de Pinieux, Marie Karanian, Francois Le Loarer, Claire Chemin-Airiau, Myriam Jean-Denis, Maud Toulmonde, Antoine Italiano, Axel Le Cesne, Nicolas Penel, Francoise Ducimetiere, Jean-Michel Coindre, Jean-Yves Blay.

**Project administration:** Gonzague de Pinieux, Marie Karanian, Francois Le Loarer, Claire Chemin-Airiau, Myriam Jean-Denis, Jean-Baptiste Courrèges, Nouria Mesli, Juliane Berchoud, Maud Toulmonde, Antoine Italiano, Axel Le Cesne, Nicolas Penel, Francoise Ducimetiere, Francois Gouin, Jean-Michel Coindre, Jean-Yves Blay.

**Resources:** Sophie Le Guellec, Sylvie Chabaud, Philippe Terrier, Corinne Bouvier, Maxime Batistella, Agnès Neuville, Yves-Marie Robin, Jean-Francois Emile, Anne Moreau, Agnes Leroux, Nathalie Stock, Marick Lae, Francoise Collin, Nicolas Weinbreck, Sebastien Aubert, Florence Mishellany, Celine Charon-Barra, Sabrina Croce, Laurent Doucet, Isabelle Quintin-Rouet, Marie-Christine Chateau, Celine Bazille, Isabelle Valo, Bruno Chetaille, Nicolas Ortonne, Anne Brouchet, Philippe Rochaix, Anne Demuret, Jean-Pierre Ghnassia, Lenaig Mescam, Nicolas Macagno, Isabelle Birtwisle-Peyrottes, Christophe Delfour, Emilie Angot, Isabelle Pommepuy, Dominique Ranchere, Claire Chemin-Airiau, Myriam Jean-Denis, Yohan Fayet, Jean-Baptiste Courrèges, Nouria Mesli, Juliane Berchoud, Francoise Ducimetiere, Francois Gouin, Jean-Michel Coindre, Jean-Yves Blay.

**Software:** Myriam Jean-Denis, Jean-Baptiste Courrèges, Nouria Mesli, Juliane Berchoud, Maud Toulmonde, Francoise Ducimetiere, Jean-Michel Coindre.

**Supervision:** Gonzague de Pinieux, Marie Karanian, Francois Le Loarer, Claire Chemin-Airiau, Myriam Jean-Denis, Yohan Fayet, Jean-Baptiste Courrèges, Nouria Mesli, Juliane Berchoud, Antoine Italiano, Axel Le Cesne, Nicolas Penel, Francoise Ducimetiere, Francois Gouin, Jean-Michel Coindre, Jean-Yves Blay.

**Validation:** Gonzague de Pinieux, Marie Karanian, Sophie Le Guellec, Sylvie Chabaud, Philippe Terrier, Corinne Bouvier, Maxime Batistella, Agnès Neuville, Yves-Marie Robin, Jean-Francois Emile, Anne Moreau, Frederique Larousserie, Agnes Leroux, Nathalie Stock, Marick Lae, Francoise Collin, Nicolas Weinbreck, Sebastien Aubert, Florence Mishellany,

Celine Charon-Barra, Sabrina Croce, Laurent Doucet, Isabelle Quintin-Rouet, Marie-Christine Chateau, Celine Bazille, Isabelle Valo, Bruno Chetaille, Nicolas Ortonne, Anne Brouchet, Philippe Rochaix, Anne Demuret, Jean-Pierre Ghnassia, Lenaig Mescam, Nicolas Macagno, Isabelle Birtwisle-Peyrottes, Christophe Delfour, Emilie Angot, Isabelle Pommepuy, Dominique Ranchere, Claire Chemin-Airiau, Myriam Jean-Denis, Jean-Baptiste Courrèges, Nouria Mesli, Juliane Berchoud, Maud Toulmonde, Antoine Italiano, Axel Le Cesne, Nicolas Penel, Francoise Ducimetiere, Francois Gouin, Jean-Michel Coindre, Jean-Yves Blay.

**Visualization:** Sophie Le Guellec, Sylvie Chabaud, Philippe Terrier, Corinne Bouvier, Maxime Batistella, Yves-Marie Robin, Jean-Francois Emile, Anne Moreau, Frederique Larousserie, Agnes Leroux, Nathalie Stock, Marick Lae, Francoise Collin, Nicolas Weinbreck, Sebastien Aubert, Florence Mishellany, Celine Charon-Barra, Sabrina Croce, Isabelle Quintin-Rouet, Marie-Christine Chateau, Celine Bazille, Isabelle Valo, Bruno Chetaille, Nicolas Ortonne, Anne Brouchet, Philippe Rochaix, Anne Demuret, Jean-Pierre Ghnassia, Lenaig Mescam, Nicolas Macagno, Isabelle Birtwisle-Peyrottes, Christophe Delfour, Emilie Angot, Isabelle Pommepuy, Dominique Ranchere, Claire Chemin-Airiau, Yohan Fayet, Jean-Baptiste Courrèges, Nouria Mesli, Juliane Berchoud, Axel Le Cesne, Francoise Ducimetiere, Jean-Michel Coindre, Jean-Yves Blay.

**Writing – original draft:** Axel Le Cesne, Francoise Ducimetiere, Jean-Yves Blay.

**Writing – review & editing:** Gonzague de Pinieux, Marie Karanian, Francois Le Loarer, Sophie Le Guellec, Sylvie Chabaud, Philippe Terrier, Corinne Bouvier, Maxime Batistella, Agnès Neuville, Yves-Marie Robin, Jean-Francois Emile, Anne Moreau, Frederique Larousserie, Agnes Leroux, Nathalie Stock, Marick Lae, Francoise Collin, Nicolas Weinbreck, Sebastien Aubert, Florence Mishellany, Celine Charon-Barra, Sabrina Croce, Laurent Doucet, Isabelle Quintin-Rouet, Marie-Christine Chateau, Celine Bazille, Isabelle Valo, Bruno Chetaille, Nicolas Ortonne, Anne Brouchet, Philippe Rochaix, Anne Demuret, Jean-Pierre Ghnassia, Lenaig Mescam, Nicolas Macagno, Isabelle Birtwisle-Peyrottes, Christophe Delfour, Emilie Angot, Isabelle Pommepuy, Dominique Ranchere, Claire Chemin-Airiau, Myriam Jean-Denis, Yohan Fayet, Jean-Baptiste Courrèges, Nouria Mesli, Juliane Berchoud, Maud Toulmonde, Antoine Italiano, Axel Le Cesne, Nicolas Penel, Francoise Ducimetiere, Francois Gouin, Jean-Michel Coindre, Jean-Yves Blay.

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
