## [Decision Letter · Decision Letter 0]

27 Jul 2020

PONE-D-20-18848

Nationwide incidence of sarcomas and connective tissue tumors of intermediate malignancy over four years using an expert pathology review network.

PLOS ONE

Dear Dr. Blay,

Thank you for submitting your manuscript to PLOS ONE. After careful consideration, we feel that it has merit but does not fully meet PLOS ONE’s publication criteria as it currently stands. Therefore, we invite you to submit a revised version of the manuscript that addresses the points raised during the review process.

The manuscript has been deemd of  high interest for the sarcoma and rare cancer community. However, several comments have been reported by reviewers and these should be thoroughly addressed.

We look forward to receiving your revised manuscript.

Kind regards,

Sandro Pasquali, M.D., Ph.D.

Academic Editor

PLOS ONE

Journal Requirements:

2. In the Methods, please clarify that participants provided oral consent. Please also state in the Methods:

- Why written consent could not be obtained

- Whether the Institutional Review Board (IRB) approved use of oral consent

- How oral consent was documented

For more information, please see our guidelines for human subjects research: https://journals.plos.org/plosone/s/submission-guidelines#loc-human-subjects-research

3.We noticed minor instances of text overlap with the following previous publication(s), which need to be addressed:

(1) https://www.journals.elsevier.com/annals-of-oncology

The text that needs to be addressed involves the Introduction section.

In your revision please ensure you cite all your sources (including your own works), and quote or rephrase any duplicated text outside the methods section. Further consideration is dependent on these concerns being addressed.

4. To comply with PLOS ONE submission guidelines, in your Methods section, please provide additional information regarding your statistical analyses, including the threshold set of statistical significance for your analyses. For more information on PLOS ONE's expectations for statistical reporting, please see https://journals.plos.org/plosone/s/submission-guidelines.#loc-statistical-reporting.

5. Please amend your list of authors on the manuscript to ensure that each author is linked to an affiliation. Authors’ affiliations should reflect the institution where the work was done (if authors moved subsequently, you can also list the new affiliation stating “current affiliation:….” as necessary).

Reviewers' comments:

Reviewer's Responses to Questions

**Comments to the Author**

1. Is the manuscript technically sound, and do the data support the conclusions?

Reviewer #1: Partly

Reviewer #2: Yes

Reviewer #3: Yes

Reviewer #4: Yes

2. Has the statistical analysis been performed appropriately and rigorously? 

Reviewer #1: Yes

Reviewer #2: Yes

Reviewer #3: Yes

Reviewer #4: No

3. Have the authors made all data underlying the findings in their manuscript fully available?

Reviewer #1: Yes

Reviewer #2: Yes

Reviewer #3: Yes

Reviewer #4: Yes

4. Is the manuscript presented in an intelligible fashion and written in standard English?

Reviewer #1: No

Reviewer #2: No

Reviewer #3: Yes

Reviewer #4: Yes

5. Review Comments to the Author

Reviewer #1: The authors provided for the first time incidence rate for specific sarcomas histology based on central pathology review. this is extremely important and useful since no high quality data on sarcoma specific histotype are available. However, the manuscript need extensive revision to be considered for publication. please refer to the specific comments below.

Introduction

Since 2013, the overall accrual in the database reached a plateau, suggesting that the closest to exhaustive collection of cases in this country was obtained. The authors should provide stronger evidence of the nationwide representativeness of the registry considering cases not accessing the network e.g. old sarcoma pts, sarcomas diagnosed only from autoptic cases etc. in France there are population based cancer registries. Although these registries are not national, comparison of data could be relevant to understand the completeness of case ascertained by NETSARC vs the population based one for malignant sarcomas only.

The RREPS/NETSARC Database

The authors should better describe the quality of the DB including the completeness of the follow-up information which are essential to define the prevalence.

Presentation of the data

The authors used the 2013 WHO classification. However the years used to estimate incidence are 2013-2016. to what extend the 2013 classification was used already in 2013 nationwide? could the author comment on any possible impact of the implementation of the 2013 during the years included in the study on the provided incidence? the authors should clarify which codes of the WHO were used to define the grouping and histological entities presented in the Tabes.

Incidence

“The NETSARC database includes 156 individual tumors or groups of sarcoma/TIM, 31 groups of sarcomas/TIM (e.g. « liposarcoma ») and 125 distinct individual histological subtypes of sarcomas or

TIM (Table 1-3)”. Can the authors better explain these different grouping reported in the results section?

Ti compare the incidence estimated in NETSARC with that of previous study, the authors should pay attention to the years included in the different study (e.g. those before 2000 may not include GIST) and the sarcomas included (e.g. population-based cancer registries study include only malignant thus TIM are excluded). The authors should re considering their comments about the comparison with previous study.

Finally, the results should separate sarcomas from TIM.

Over 100-fold difference in incidence in different sarcoma histotypes

Figure 1 presents the individual histotypes and relevant groups of histotypes (eg liposarcoma, leiomyosarcoma, unterine sarcomas) ordered by incidence. I think there is a mistake, the author mean Table 1.

The author should use coherent terms/label in the text and in Table 1.

There were respectively 35, 63 and 66 different histological subtypes or groups (e.g. MPNST, or

vascular sarcomas…) of sarcomas or TIM with an incidence ranging from 10 to 1/10 6 /year, 1-0.1/10 6

per year, or <0.1/10 6 /year respectively. Please, clarify how to find these numbers in Table 1.

the description of the results is very confusing and it does not provide a guideline to read Table 1. Please revise

Table 2, 3 and 4 are not presented in the results. Sex ratio and mean age for diagnosis should be commented in the results section.

Table 1

The author should clarify what it is included in table 1. number by year and incidence rate in the 4 years of the study? in red are marked ....? in black bold are marked....?

Table 2

The author should clarify what it is included in table 2. number by year and incidence rate in the 4 years of the study? in black bold are marked....?

Table 3

The author should clarify what it is included in table 3. number by year and incidence rate in the 4 years of the study? in red are marked ....? in black bold are marked....?

the authors should also consider to present the incidence rate by decreasing rate?

Figure 1

can the authors clarify the colours in figure 1?

the text needs English revision.

Reviewer #2: TITLE: Nationwide incidence of sarcomas and connective tissue tumors of intermediate malignancy over four years using an expert pathology review network.

MAIN CONCERN

(1)

Using a nationwide database, this important study aims to describe the incidence of fully malignant sarcomas and mesenchymal tumors of intermediate malignancy (TIM), all with expert confirmation of pathologic diagnosis. Although appearing in the title, the term “connective tissue tumors of intermediate malignancy” was never explained/clarified in the manuscript. The criteria for differential diagnosis were never reported.

TIM cases were 6460 as can be read on line 11 of Abstract. However, those reporting “intermediate” in the name of their histotype were only 63 cases (table 1). In particular they were: 5 cases of “Intermediate fibrohistiocytic tumors”; 6 cases of “Intermediate vascular tumors”; and 52 cases of “Sarcoma NOS Tumors of intermediate malignancy”. The difference between 6460 and 63 was never apprised in the article.

In the analysis, the tumors were broken down in two-way tables in which rows were always the histological subtypes and columns were time or person characteristics. There were no separate tables for fully malignant sarcomas and TIMs. They were reported as sum at each cell of row and column interception; the reader cannot know their frequency separately.

In my opinion, if the tables cannot be changed, the title should be rephrased.

(2)

The manuscript needs English editing. Moreover, I strongly suggest to also control all numbers appearing in the text. For example: “139 histological subtypes” (Abstract, line 10) should be “159 histological subtypes” that is the sum of “30, 63 and 66 different histological subtypes of sarcomas or TIM”, reported on line 15-16 of Abstract. Another example: the percentage of sarcomas (=18710/25172) is 74%, not 64%” as it appears in line 10 of Abstract. On line 11 the percentage of TIMs should be 26% rather that 36%.

MINOR CONCERN

(1)

In Table 4, the heading of the second column is “F/H” (Femmes/Hommes in French) rather than F/M (Females/Males).

Reviewer #3: No Major Critiques. The manuscript is largely clear, informative, and provides the most detailed breakdown of sarcoma type incidence in a population that I have seen to date.

Minor:

1. The authors should carefully check the manuscript for a few scattered English grammar issues and typos (both in text and tables), and use more consistent capitalization in the tables (some tumors have only the first word of the name capitalized, others have multiple words capitalized)

2. I would not necessarily consider adenosarcoma or phyllodes, much less UT resembling ovarian sex cord stromal tumor as proper sarcomas, though I do appreciate why they were included here. The authors may wish to clarify if the category in table 1 part 2 refers specifically to malignant phyllodes with sarcomatous overgrowth or all malignant phyllodes.

3. In table 2 does the “osteosarcoma” entry include the subtypes or is it separate – if separate how is it different than the NOS category. There are 2 entries for low grade central osteosarcoma with different numbers of cases. This table should be checked for accuracy and clarity

4. Figure 1 looks obviously pasted from a spreadsheet (including some cut off text visible in the top of the middle and right panels) and is hard to read as presented. Would consider removing tumors with no published clinical trials to a supplemental table. Or else presenting each subset broken down by incidence into individual figures for improved legibility. Why is synovial sarcoma NOS separated from monophasic SS and biphasic SS? Or MPNST usual type from MPNST? Or SFT (all) from SFT NOS? Consider combining or eliminating duplicate /redundant entries as these do not seem to add much to the table or understanding of trial availability. At a minimum trials involving specific variants of individual sarcoma types should be grouped together for clarity.

5. In the discussion specify the WHO classification used (obviously these had to have been done using the 2013 WHO given when the data was collected) but it is no longer “the most recent”.

Reviewer #4: General Comments

This work is an important report describing the sarcoma incidence in France, where an efficient system of cases centralization is in place. This work will be the benchmark for future epidemiological studies.

The manuscript however has two major critical points:

1) the data are displayed in long tables; If the presented data were displayed in a more graphically appealing way the work would gain a lot; after all the main point of the present manuscript is to convey the numbers, so the data visualization probably would represent the analytical part of the paper.

2) Statistical analysis de-trending the sarcoma count among different histotypes is arguable; however this do not diminish the value of this work, that relies in the numbers provided. see the specific comment for explanation.

Specific comments

Abstract

Many readers are probably not aware of what NETSARC and RREPS are; a more descriptive term could help the reader of the abstract and push him/her to see the paper.

Introduction

Data from years 2010-12 have been dropped from further analysis, probably a plot showing the total number of cases might help the reader to picture the story you are telling.

Materials and methods

how the central review is enforced?

Clinical trials

it is not clear to me how the data have been retrieved, and tabulated.

Statistics

The importance of the presented data is enough to do not require a statistical analysis. However I think that the proposed analysis of trend is not adequate: the analysis of variance should compare the variance of a variable among 2 or more groups; this variable should be normally distributed, but the variable of interest is a count and in this case normality is not a good assumption, since the numbers are really low given the rarity of sarcoma (a Poisson distribution might have the right characteristics). Moreover there is a single data point that cannot show "variance"; I would therefore take the statistical significance with a lots of doubts. On the other hand the figure shown in supplementals are linear models; probably better, they in fact treat time as a continuous variable, however they also expect negative count to be perfectly normal.

If the authors really think that de-trend a time series with 4 data point is an important aim for their work I suggest to use a statistical procedure that does the job (as a spline or a gaussian process), and – given the limited number of observation between the many different categories – a regularizing technique or a fully bayesian approach with multilevel models might help.

lastly the data shown in the table next to the supplementary figure has a error: the line copied for central chondrosarcoma is not correct neither would give the result reported ( I suspect the two categories of grading have been inverter).

Results

There are many typos; also tables need revision (commas instead of points, H = Homes ).

see statistics

Discussion

Also typos; "most recent WHO" is the 2020?

see statistics

Figure 1: is not actually a figure. the same general comment on data representation apply here

6. PLOS authors have the option to publish the peer review history of their article (what does this mean?). If published, this will include your full peer review and any attached files.

Reviewer #1: No

Reviewer #2: **Yes: **Giuseppe Mastrangelo

Reviewer #3: No

Reviewer #4: **Yes: **Salvatore Lorenzo Renne

---

## [Author Response · Author response to Decision Letter 0]

15 Sep 2020

JY Blay

Jean-yves.blay@lyon.unicancer.fr

to

Sandro Pasquali, M.D., Ph.D.

Academic Editor

PLOS ONE

PONE-D-20-18848 Nationwide incidence of sarcomas and connective tissue tumors of intermediate malignancy over four years using an expert pathology review network.

Dear Dr. Pasquali,

Thank you for your recent e-mail regarding the above mentioned article submitted to PLOS ONE. 

We are now able to submit a revised version of this article taking into consideration all rthe requirements of the reviewers. 

Please find hereunder a point by point reply to the different comments. 

A revised version with visible marks in included as well as the revised version. 

We remain of course at your disposition for any further question you may have regarding the work

With kind regards

 JY Blay

 

Journal Requirements:

This was done accordingly. We have inserted the table at the end though to facilitate legibility. As mentioned by one of the 4 reviewers, we could split the tables (e.g. Table 1 in Table 1A, 1B etc..) if preferred by the editorial board of PLoS One

2. In the Methods, please clarify that participants provided oral consent. Please also state in the Methods:

We confirm that all patients included in this work approved orally their participation to the pathology review, and the inscription in the national network database according to the national laws at that time, as well as the recommendations of the French National Cancer Institute (page 6 revised version).

- Why written consent could not be obtained.

The mission given by the French National Cancer institute (INCA)was actually to review all pathology samples and to collect a minimal set of anonymized data to be able to monitor patient outcome. These were not clinical trials therefore. It was a national mission given to the networks for the management of patients with sarcomas. However, as indicated in the document and in previous, not all patients did actually had their samples reviewed by reference centers immediately in the first years (p 6 of the revised Ms).

- Whether the Institutional Review Board (IRB) approved use of oral consent

The activity of the networks had not to be reviewed by the national ethics committees as done for clinical studies (the Comité de Protection des Personnes). The local hospitals (ie the reference centers) use their own internal procedure (internal institutional review board) for this approval, which was mandatory for the participation to the networks (p6 of the revised Ms).

- How oral consent was documented

Standard information sheet is given to all patients in each institution at first entry, with the opportunity for the patient to opt-out participation. In addition, review of the patient records in indicated in all multidisciplinary sarcoma boards reports. This is also indicated on p6 of the revised Ms.

For more information, please see our guidelines for human subjects research: https://smex-ctp.trendmicro.com:443/wis/clicktime/v1/query?url=https%3a%2f%2fjournals.plos.org%2fplosone%2fs%2fsubmission%2dguidelines%23loc%2dhuman%2dsubjects%2dresearch&umid=59435c4f-7e81-48ac-8a14-c7d99c29f380&auth=88b027512158546030ba1422e012c48ffde03353-fec0bc3097bbb29ab85c8bc38a232be8d0801f9tumor board 0

 3.We noticed minor instances of text overlap with the following previous publication(s), which need to be addressed:

(1) https://www.journals.elsevier.com/annals-of-oncology

Please accept our apologizes for this. We, as not native english speakers, often use similar sentences for the same topics, in particular when the same data set is used for different questions. We have modified the text in the revised accordingly (in particular in the introduction and discussion).

The text that needs to be addressed involves the Introduction section. 

The introduction was modified.as indicated in the above comment

 In your revision please ensure you cite all your sources (including your own works), and quote or rephrase any duplicated text outside the methods section. Further consideration is dependent on these concerns being addressed.

This was carefully reviewed and done the different sections of these aspects.

4. To comply with PLOS ONE submission guidelines, in your Methods section, please provide additional information regarding your statistical analyses, including the threshold set of statistical significance.

Thresholds for significance were added on p7 of the revised M/M section.

5. Please amend your list of authors on the manuscript to ensure that each author is linked to an affiliation. Authors’ affiliations should reflect the institution where the work was done (if authors moved subsequently, you can also list the new affiliation stating “current affiliation:….” as necessary).

This was done on the Ms and on the website. Please note that some of the reference centers include patients from more than one hospital, to match the organizational requests of the French INCA.

Captions for supplementary data were added.

 

Replies to reviewers' comments:

Reviewer's Responses to Questions

 Comments to the Author

1. Is the manuscript technically sound, and do the data support the conclusions?

 Reviewer #1: Partly

Reviewer #2: Yes

Reviewer #3: Yes

Reviewer #4: Yes

2. Has the statistical analysis been performed appropriately and rigorously? 

Reviewer #1: Yes

Reviewer #2: Yes

Reviewer #3: Yes

Reviewer #4: No

3. Have the authors made all data underlying the findings in their manuscript fully available?

Reviewer #1: Yes

Reviewer #2: Yes

Reviewer #3: Yes

Reviewer #4: Yes

4. Is the manuscript presented in an intelligible fashion and written in standard English?

Reviewer #1: No

Reviewer #2: No

Reviewer #3: Yes

Reviewer #4: Yes

5. Review Comments to the Author

Reviewer #1: The authors provided for the first time incidence rate for specific sarcomas histology based on central pathology review. this is extremely important and useful since no high quality data on sarcoma specific histotype are available. However, the manuscript need extensive revision to be considered for publication. please refer to the specific comments below.

Introduction

Since 2013, the overall accrual in the database reached a plateau, suggesting that the closest to exhaustive collection of cases in this country was obtained. The authors should provide stronger evidence of the nationwide representativeness of the registry considering cases not accessing the network e.g. old sarcoma pts, sarcomas diagnosed only from autoptic cases etc. in France there are population based cancer registries. Although these registries are not national, comparison of data could be relevant to understand the completeness of case ascertained by NETSARC vs the population based one for malignant sarcomas only.

This is indeed a very important question. As rightly pointed out by the reviewer, there are no national registries for any cancers in France. There are several limitations with the use of regional registries related to 1) the 20-30% rate of histotype modification after pathology review by expert centers in sarcoma (reported by several works, reference in tis article), and 2) the lack of exhaustivity of these regional registries for sarcomas. An ongoing work, yet unpublished, where NETSARC contributed indicates that 65-70% of incident cases as calculated in the work are declared in these registries, pointing to the need of an increased collaboration between these structures. This has been added in the discussion section of the revised Ms, p10. 

The RREPS/NETSARC Database

The authors should better describe the quality of the DB including the completeness of the follow-up information which are essential to define the prevalence.

This is again an important question. The database is completed and monitored by a group of dedicated CRAs (as for a clinical trial) for baseline information, clinical presentation; outcome… The majority of the cases are obtained directly from the pathologist laboratories, but a fraction of the patients are first seen by the clinical MDT, and then referred to the expert pathology laboratory for confirmation of the diagnosis. This double source of entry contributes to improve the exhaustivity of the collection of cases. This has been added in the Material and method section p7.

Presentation of the data

The authors used the 2013 WHO classification. However, the years used to estimate incidence are 2013-2016. to what extend the 2013 classification was used already in 2013 nationwide? could the author comment on any possible impact of the implementation of the 2013 during the years included in the study on the provided incidence? the authors should clarify which codes of the WHO were used to define the grouping and histological entities presented in the Tables.

The 2013 classification was used from the initiation of this work indeed, as added in the M/M section p7. Monthly physical meetings of the pathologists network for review of complex cases have facilitated the homogeneity of data collection within this group. The addition of the WHO codes is a challenge for the legibility of the tables already quite crowded. We respectfully suggest to keep this presentation on this simpler format.

Incidence

“The NETSARC database includes 156 individual tumors or groups of sarcoma/TIM, 31 groups of sarcomas/TIM (e.g. « liposarcoma ») and 125 distinct individual histological subtypes of sarcomas or

TIM (Table 1-3)”. Can the authors better explain these different grouping reported in the results section?

Indeed, this was an important difficulty of the presentation of this large and fragmented dataset. We considered that it was still useful to present each individual histotype, (e.g. WDLPS) and groups of histotypes when clinically relevant (e.g. WD and DDLPS, or even liposarcoma as a whole). Conversely, when a grouping was not clinically meaningful in clinical routine (e.g. “fibroblastic and myofibroblastic tumors” in table 1) we did not consider this a a distinct entity. This point was added in the M/M (p7) and in the Result section of the revised document (p8). We have reformulated these section and provide some examples to improve clarity. In order to avoid a debate on the exact number of histotypes which can be subjective, we chose to present tthat the total number of histotypes was above 150.

To compare the incidence estimated in NETSARC with that of previous study, the authors should pay attention to the years included in the different study (e.g. those before 2000 may not include GIST) and the sarcomas included (e.g. population-based cancer registries study include only malignant thus TIM are excluded). The authors should re considering their comments about the comparison with previous study.

We fully agree with this observation. A note of caution has been added in the discussion section (p10, “It is however important to point hat comparison with historical series have limitations also because individual entities may have only a recent “existence”, (e.g. GIST before 1999, solitary fibrous tumors , etc…). The numbers presented are those measured with a more recent classification which has evolved over the years”.

Finally, the results should separate sarcomas from TIM. Over 100-fold difference in incidence in different sarcoma histotypes Figure 1 presents the individual histotypes and relevant groups of histotypes (eg liposarcoma, leiomyosarcoma, unterine sarcomas) ordered by incidence. I think there is a mistake, the author mean Table 1. 

We actually meant both table 1 and Figure 1, where histotypes are presented ordered by incidence. This was corrected in the revised version (p9).

The author should use coherent terms/label in the text and in Table 1.

There were respectively 35, 63 and 66 different histological subtypes or groups (e.g. MPNST, or vascular sarcomas…) of sarcomas or TIM with an incidence ranging from 10 to 1/10 6 /year, 1-0.1/10 6 per year, or <0.1/10 6 /year respectively. Please, clarify how to find these numbers in Table 1. the description of the results is very confusing and it does not provide a guideline to read Table 1. Please revise

We agree that this presentation was too complicated. We removed this sentence to improve legibility on p8, and reformulated section 3 of the Revised Result section. Note that there are 5 groups of histotypes with an incidence >10/10 6, 30 with an incidence 1-10/10 6, 64 (not 63) with and incidence from 1 to 0.1/10 6, and 66 with an incidence <0.1/10 6.

Table 2, 3 and 4 are not presented in the results. Sex ratio and mean age for diagnosis should be commented in the results section.

We agree that the previous presentation was unclear (Table 1-3”). In the revised version we added more explicitly Table 1, 2 and 3 quote in the text (first paragraph), as well as Table 4 (last section) on p8 and p9.

Table 1

The author should clarify what it is included in table 1. number by year and incidence rate in the 4 years of the study? in red are marked ....? in black bold are marked....?

This was indeed not enough described. In red are indicated the groups of histological subtypes. In bold the subgroups of histological subtypes. We also added “Leiomyosarcoma (all)”, “MPNST (all)” in table 1, “Osteosarcoma (all)”, “Chondrosarcoma (all)” in Table 2. We made the color code and bold homogenous on the first three tables. We added this information in the revised results section.

Table 2

The author should clarify what it is included in table 2. number by year and incidence rate in the 4 years of the study? in black bold are marked....?

As mentioned above indeed : this was indeed not enough described. We added this information in the revised results section

Table 3

The author should clarify what it is included in table 3. number by year and incidence rate in the 4 years of the study? in red are marked ....? in black bold are marked....?

As mentioned above indeed : this was indeed not enough described. We added this information in the revised results section

the authors should also consider to present the incidence rate by decreasing rate?

It is our intent to present it this way in figure 1. We tried to clarify this is the revised version

Figure 1

can the authors clarify the colours in figure 1?

The colours were added to improve a global graphic legibility of the availability of published clinical trials according to the incidence of the histotype. But indeed, this was not clear enough. We added this in the revised version in the legend of the figure.

the text needs English revision.

We have had reviewed the text accordingly by an English native speaker.

Reviewer #2: TITLE: Nationwide incidence of sarcomas and connective tissue tumors of intermediate malignancy over four years using an expert pathology review network.

MAIN CONCERN

(1)

Using a nationwide database, this important study aims to describe the incidence of fully malignant sarcomas and mesenchymal tumors of intermediate malignancy (TIM), all with expert confirmation of pathologic diagnosis. Although appearing in the title, the term “connective tissue tumors of intermediate malignancy” was never explained/clarified in the manuscript. The criteria for differential diagnosis were never reported.

TIM cases were 6460 as can be read on line 11 of Abstract. However, those reporting “intermediate” in the name of their histotype were only 63 cases (table 1). In particular they were: 5 cases of “Intermediate fibrohistiocytic tumors”; 6 cases of “Intermediate vascular tumors”; and 52 cases of “Sarcoma NOS Tumors of intermediate malignancy”. The difference between 6460 and 63 was never apprised in the article.

We agree that this was not clear enough in the previous version. Tumor of intermediate malignancy (TIM) designation is attributed according to the WHO 2013 classification (ref 1 of the Ms), e.g. atypical lipomatous tumors, dermatofibrosarcoma protuberans, fibromatosis, hemangioendothelioma, giant cell tumor of the bone to quote a few (also summarized in (http://sarcomahelp.org/reviews/who-classification-sarcomas.html). The term intermediate is not necessarily including in the histological designation in many subtypes. This has been added in the material and method section of the revise Ms on p7.

In the analysis, the tumors were broken down in two-way tables in which rows were always the histological subtypes and columns were time or person characteristics. There were no separate tables for fully malignant sarcomas and TIMs. They were reported as sum at each cell of row and column interception; the reader cannot know their frequency separately.

In my opinion, if the tables cannot be changed, the title should be rephrased.

We agree with this recommendation and made the corrections in the titles and in the captions. As discussed before, we chose to present the rows according to the WHO classification which is based on histological differentiation mainly, adding few specific points on the primary site of the tumors. For this reason, we adapted and developed the titles and legend of Table 1, 2, 3 accordingly rather than to split TIM and Sarcomas in different tables. 

(2)

The manuscript needs English editing. Moreover, I strongly suggest to also control all numbers appearing in the text. For example: “139 histological subtypes” (Abstract, line 10) should be “159 histological subtypes” that is the sum of “30, 63 and 66 different histological subtypes of sarcomas or TIM”, reported on line 15-16 of Abstract. Another example: the percentage of sarcomas (=18710/25172) is 74%, not 64%” as it appears in line 10 of Abstract. On line 11 the percentage of TIMs should be 26% rather that 36%.

We have made corrections in the numbers indicated in the abstract which were indeed (for one) partly different from those shown in the figure. We also agree that the previous version was unclear for several of the numbers, mostly because groups of diseases (e.g. liposarcoma, notwithstanding molecular subtype) are frequently considered in publications e.g. for clinical trials (see above). One of the challenges of these numbers is that we are both indicating 1) individual histotypes, and 2) pooled histotypes in groups (e.g. liposarcomas). We have amended the text, abstract and discussion to make this important point clearer, and for instance changed the first sentence of the abstract results to be inclusive on the number of patients. We have been working with an english speaking person for the revised version. We have reviewed and homogenized the presentation of the numbers of histotypes in the revised version all along the text as suggested by this reviewer. 

MINOR CONCERN

(1)

In Table 4, the heading of the second column is “F/H” (Femmes/Hommes in French) rather than F/M (Females/Males).

We have corrected this typographic error.

Reviewer #3: No Major Critiques. The manuscript is largely clear, informative, and provides the most detailed breakdown of sarcoma type incidence in a population that I have seen to date.

Minor:

1. The authors should carefully check the manuscript for a few scattered English grammar issues and typos (both in text and tables), and use more consistent capitalization in the tables (some tumors have only the first word of the name capitalized, others have multiple words capitalized)

We have corrected the revised manuscript to be consistent on these aspects

2. I would not necessarily consider adenosarcoma or phyllodes, much less UT resembling ovarian sex cord stromal tumor as proper sarcomas, though I do appreciate why they were included here. The authors may wish to clarify if the category in table 1 part 2 refers specifically to malignant phyllodes with sarcomatous overgrowth or all malignant phyllodes.

We agree and have modified the tables and legends accordingly 

3. In table 2 does the “osteosarcoma” entry include the subtypes or is it separate – if separate how is it different than the NOS category. There are 2 entries for low grade central osteosarcoma with different numbers of cases. This table should be checked for accuracy and clarity

We have reviewed the number and confirmed the different numbers. In the case of osteosarcoma, indeed the reviewers indicated “Osteosarcoma” without further precision, or “Osteosarcoma NOS”. though we could have pooled them into “Osteosarcoma NOS” we decided not to do so to describe as accurately as possible what was added by the reviewers. However, for clarity we added a line summing all osteosarcomas to have a broader picture. We added this comment in the legend of the table (p14). 

4. Figure 1 looks obviously pasted from a spreadsheet (including some cut off text visible in the top of the middle and right panels) and is hard to read as presented. Would consider removing tumors with no published clinical trials to a supplemental table. Or else presenting each subset broken down by incidence into individual figures for improved legibility. Why is synovial sarcoma NOS separated from monophasic SS and biphasic SS? Or MPNST usual type from MPNST? Or SFT (all) from SFT NOS? Consider combining or eliminating duplicate /redundant entries as these do not seem to add much to the table or understanding of trial availability. At a minimum trials involving specific variants of individual sarcoma types should be grouped together for clarity.

It is indeed very challenging to present clinical trials in this section in a graphical and legible format. Our intent with the colour codes was to show that each individual subtype of sarcoma was not less likely associated with a clinical trial as they were rarer. Some very rare histotypes (infantile fibrosarcoma, or ESS for instance) are however specifically covered by a clinical trial. In the examples chosen by the reviewer, there are trials for synovial sarcomas, but not for biphasic or monophasic synovial sarcoma specifically. Respectfully, we suggest to develop these points in the legend of the table rather than to change the structure of the table. These changes were made in the result, and discussion section section (p11) and in the legend of the figure in the revised version (p14). If preferred by the editor, we of course would agree to put this figure in the supplemental figures.

5. In the discussion specify the WHO classification used (obviously these had to have been done using the 2013 WHO given when the data was collected) but it is no longer “the most recent”.

We agree we corrected this point in the different section of the document (M/M, discussion)

Reviewer #4: General Comments

This work is an important report describing the sarcoma incidence in France, where an efficient system of cases centralization is in place. This work will be the benchmark for future epidemiological studies.

The manuscript however has two major critical points:

1) the data are displayed in long tables; If the presented data were displayed in a more graphically appealing way the work would gain a lot; after all the main point of the present manuscript is to convey the numbers, so the data visualization probably would represent the analytical part of the paper.

We agree with this reviewer that this is indeed a challenge for the presentation. We would agree of course to split the tables, with e.g. Table 1A,1B etc for each individual group of disease if requested and approved by the editors of PLOS one. 

2) Statistical analysis de-trending the sarcoma count among different histotypes is arguable; however this do not diminish the value of this work, that relies in the numbers provided. see the specific comment for explanation.

We agree with this commentary. A note of caution was added in the discussion section of the revised version (see under).

Specific comments

Abstract

Many readers are probably not aware of what NETSARC and RREPS are; a more descriptive term could help the reader of the abstract and push him/her to see the paper.

We agree that a more precise description is needed this is added in the revised version in particular in the material and method section (p7).

Introduction

Data from years 2010-12 have been dropped from further analysis, probably a plot showing the total number of cases might help the reader to picture the story you are telling.

We have added the numbers of patients seen from 2010 to 2012 in the revised text, in the result section, briefly. We feared that presenting these data in table would increase the complexity of this presentation

Materials and methods

how the central review is enforced?

There are several mechanisms : 1) each pathology review report was and is indicating the mention of the mandatory review, the sites and contact of the expert pathologists and all french pathologist receive it multiple times per year; 2) when missed, patients presented in clinical MDT without pathology review are immediately referred to the closest expert pathology center ; 3) the patient themselves exerted on multiple occasion their request for a pathology review , with the dissemination on the internet of the existence of networks of excellence and the mandatory pathology and clinical review. These different points are added in the revised version of the Ms (p10).

Clinical trials

it is not clear to me how the data have been retrieved, and tabulated.

This came from an exhaustive review of published literature using Pubmed. This was further precised in the Revised Material and method section (p7)

Statistics

The importance of the presented data is enough to do not require a statistical analysis. However I think that the proposed analysis of trend is not adequate: the analysis of variance should compare the variance of a variable among 2 or more groups; this variable should be normally distributed, but the variable of interest is a count and in this case normality is not a good assumption, since the numbers are really low given the rarity of sarcoma (a Poisson distribution might have the right characteristics). Moreover there is a single data point that cannot show "variance"; I would therefore take the statistical significance with a lots of doubts. On the other hand the figure shown in supplementals are linear models; probably better, they in fact treat time as a continuous variable, however they also expect negative count to be perfectly normal.

If the authors really think that de-trend a time series with 4 data point is an important aim for their work I suggest to use a statistical procedure that does the job (as a spline or a gaussian process), and – given the limited number of observation between the many different categories – a regularizing technique or a fully bayesian approach with multilevel models might help.

We agree with these proposals and have proposed an amended presentation and statistical analysis in the revised version for this additional figure, with modifications in the material and method section, in the result section and in the supplementary figure. The p value was fitted by a Poisson regression in the revised version. The histological subtypes with a significant (p<0.01) variation are adenosarcoma, desmoid, UPS, high grade ESS, pecoma, and myoepithelioma

lastly the data shown in the table next to the supplementary figure has a error: the line copied for central chondrosarcoma is not correct neither would give the result reported ( I suspect the two categories of grading have been inverter).

This is indeed a mistake. This subset was not retained by the new analysis, with the threshold used. This has been corrected in the revised manuscript.

Results

There are many typos; also tables need revision (commas instead of points, H = Homes ).

see statistics

Discussion

Also typos; "most recent WHO" is the 2020?

see statistics

We made the correction requested in this section

Figure 1: is not actually a figure. the same general comment on data representation apply here

Indeed, Figure 1 is more a table with colors to show graphically the lower number of clinical trials in the three lists of sarcoma with different incidences (100-fold difference between the 3 columns). As mentioned above, it is indeed very challenging to present clinical trials in this section in a graphical and legible format. Our intent with the color codes was to show that each individual subtype of sarcoma was not less likely associated with a clinical trial as they were rarer. In addition, in the examples chosen by the reviewer, there are trials for synovial sarcomas, but not for biphasic or monophasic synovial sarcoma specifically. Respectfully, we suggest to develop these points in the legend of the table rather than to change the structure of the Figure. These changes were made in the result section and in the legend of the figure in the revised version.

6. PLOS authors have the option to publish the peer review history of their article (what does this mean?). If published, this will include your full peer review and any attached files.

Do you want your identity to be public for this peer review? For information about this choice, including consent withdrawal, please see our Privacy Policy.

Reviewer #1: No

Reviewer #2: Yes: Giuseppe Mastrangelo

Reviewer #3: No

Reviewer #4: Yes: Salvatore Lorenzo Renne

---

## [Decision Letter · Decision Letter 1]

28 Oct 2020

PONE-D-20-18848R1

Nationwide incidence of sarcomas and connective tissue tumors of intermediate malignancy over four years using an expert pathology review network.

PLOS ONE

Dear Dr. Blay,

Thank you for submitting your manuscript to PLOS ONE. After careful consideration, we feel that it has merit but does not fully meet PLOS ONE’s publication criteria as it currently stands. Therefore, we invite you to submit a revised version of the manuscript that addresses the points raised during the review process.

The manuscript has been clearly improved. However, please carefully address further comments from reviewer 1 and minor additional comments  from reviewer 2.

We look forward to receiving your revised manuscript.

Kind regards,

Sandro Pasquali, M.D., Ph.D.

Academic Editor

PLOS ONE

Reviewers' comments:

Reviewer's Responses to Questions

**Comments to the Author**

1. If the authors have adequately addressed your comments raised in a previous round of review and you feel that this manuscript is now acceptable for publication, you may indicate that here to bypass the “Comments to the Author” section, enter your conflict of interest statement in the “Confidential to Editor” section, and submit your "Accept" recommendation.

Reviewer #1: (No Response)

Reviewer #2: All comments have been addressed

Reviewer #4: All comments have been addressed

2. Is the manuscript technically sound, and do the data support the conclusions?

Reviewer #1: Yes

Reviewer #2: Partly

Reviewer #4: Yes

3. Has the statistical analysis been performed appropriately and rigorously? 

Reviewer #1: Yes

Reviewer #2: Yes

Reviewer #4: Yes

4. Have the authors made all data underlying the findings in their manuscript fully available?

Reviewer #1: No

Reviewer #2: Yes

Reviewer #4: Yes

5. Is the manuscript presented in an intelligible fashion and written in standard English?

Reviewer #1: Yes

Reviewer #2: No

Reviewer #4: Yes

6. Review Comments to the Author

Reviewer #1: Thank you for the revised version of the paper which has greatly improved. However, some issues remain incomplete or to clarify.

Abstract

conclusion: ... and that tumors with an incidence<106 /year have a much lower access to clinical trials. I think you mean with an incidence <1/106

The RREPS/NETSARC Database

The authors should better describe the quality of the DB including the completeness of the follow-up information which are essential to define the prevalence. Could the authors expand how they ensure the completeness of the follow-up? how life status information are collected? is there any indicator of lost to follow-up or any other?

Presentation of the data

I do agree that the current table should not include the WHO codes. However, this is a key part of the material and methods selection since an article should include all the information to replicate the study. Please add the codes as supplementary Table.

“The number of patients for each individual histological subtype of sarcoma or TIM per year, from 2013 to 2016, is therefore presented in these tables. To facilitate the comparison with other databases using previous classifications, the incidence for groups of tumors are also presented in the tables, when they are clinically relevant (e.g. uterine sarcoma). We also considered that it was still useful to present each individual histotype, (e.g. WDLPS) and groups of histotypes when clinically relevant (e.g. WD and DDLPS, or even liposarcoma, leiomyosarcomas). Conversely, when a grouping was not clinically meaningful in clinical routine (e.g. “fibroblastic and myofibroblastic tumors” in table 1) we did not consider this a distinct entity.” This is still unclear. The authors should clarify what they count as a single entity. Please add a column on the left hand site of Table 1 and Table 2 indicating what was counted as single entity (N=150). It is not clear also why some grouping have the incidence and others not. it does not seem to depend on the clinical relevance because fibroblastic and myofibroblastic tumours have incidence data in Table 1. Kindly clarify.

Incidence

It is a pity not be able to separate TIM from malignant sarcomas. The authors used the WHO classifications which provides information about the sarcoma behaviour. Thus, the authors should clarify why it is not possible to distinguish TIM and malignant sarcomas. This goes back to the issues of understanding the codes used for the analyses.

Incidence of individual histotypes and published clinical trials.

“..... 14 of 35 (40%) histotypes with an incidence >1/10 6 had a dedicated phase III study vs 6 of 130 (4.6%) histotypes for sarcomas with a incidence <1/10 6 (p<10 -6 ). 20100 (79,7%) patients of the database had a specific histotype for which no phase III trial had been reported. 21 of 35 (60%) histotypes with an incidence >1/10 6 had a dedicated randomized phase II study vs 10 of 129 (7.7%) histotypes for sarcomas with a incidence <1/10 6 (p<10 -10 ))” The authors should clarify whether the histotypes for sarcomas with a incidence <1/106 are 129 or 130.

Table 1. please check the incidence for Embryonal RMS and Myoepithelioma, myoepithelial carcinoma, & mixed tumour. The incidence of the histotype sum up differently from the incidence reported.

Reviewer #2: Review Comments to the Author

Please use the space provided to explain your answers to the questions above. You may also include additional comments for the author, including concerns about dual publication, research ethics, or publication ethics. (Please upload your review as an attachment if it exceeds 20,000 characters) (Limit 100 to 20000 Characters)

Please see my comments in the attached file

Reviewer #4: The revised version addressed the points risen.

This work will represent the benchmark for future epidemiological studies.

7. PLOS authors have the option to publish the peer review history of their article (what does this mean?). If published, this will include your full peer review and any attached files.

Reviewer #1: No

Reviewer #2: **Yes: **Giuseppe Mastrangelo

Reviewer #4: **Yes: **Salvatore Lorenzo Renne

---

## [Author Response · Author response to Decision Letter 1]

6 Dec 2020

JY Blay

Jean-yves.blay@lyon.unicancer.fr

to

Sandro Pasquali, M.D., Ph.D.

Academic Editor

PLOS ONE

PONE-D-20-18848R1 Nationwide incidence of sarcomas and connective tissue tumors of intermediate malignancy over four years using an expert pathology review network.

6 Dec 2020

Dear Dr. Pasquali,

Thank you for your recent e-mail regarding the above mentioned article submitted to PLOS ONE. 

We have carefully reviewed the comments of the reviewers and are now able to submit a revised version of this article taking into consideration the requirements of the reviewers. 

Please find hereunder a point by point reply to the different comments. 

A revised version with visible marks in included as well as the revised version. 

We remain of course at your disposition for any further question you may have regarding the work

With kind regards

 JY Blay

 

PONE-D-20-18848R1

Nationwide incidence of sarcomas and connective tissue tumors of intermediate malignancy over four years using an expert pathology review network.

PLOS ONE

Dear Dr. Blay,

Thank you for submitting your manuscript to PLOS ONE. After careful consideration, we feel that it has merit but does not fully meet PLOS ONE’s publication criteria as it currently stands. Therefore, we invite you to submit a revised version of the manuscript that addresses the points raised during the review process.

The manuscript has been clearly improved. However, please carefully address further comments from reviewer 1 and minor additional comments from reviewer 2.

We look forward to receiving your revised manuscript.

Kind regards,

Sandro Pasquali, M.D., Ph.D.

Academic Editor

PLOS ONE

 Review Comments to the Author

Reviewer #1: Thank you for the revised version of the paper which has greatly improved. However, some issues remain incomplete or to clarify.

Abstract : Conclusion: ... and that tumors with an incidence<106 /year have a much lower access to clinical trials. I think you mean with an incidence <1/106

Answer The reviewer is absolutely right. We have made the appropriate modifications in the revised version

The RREPS/NETSARC Database

The authors should better describe the quality of the DB including the completeness of the follow-up information which are essential to define the prevalence. Could the authors expand how they ensure the completeness of the follow-up? how life status information are collected? is there any indicator of lost to follow-up or any other?

Answer : This is indeed an important point. The database is not systematically completed for follow-up by the CRA, as it would be in a clinical trial, but all baseline and first therapeutic informations are completed until the end of the first line treatment. This includes alll pathology reviews, which are therefore as presented , the final diagnoses. The median follow-up of the series was 17 months in the recent publications of the same dataset (25). Importantly, since 2019, the information (survival and all treatments) from the nationwide database of the social security system (SNDS, https://www.snds.gouv.fr/SNDS/Accueil ) is used to update the latest survival information of these patients as part of the Health datahub Deepsarc project (https://www.health-data-hub.fr/outil-de-visualisation), now ensuring a more exhaustive follow-up information. These points were added in the revised R2 version of the Ms.

Presentation of the data

I do agree that the current table should not include the WHO codes. However, this is a key part of the material and methods selection since an article should include all the information to replicate the study. Please add the codes as supplementary Table.

Answer : We added the WHO codes (2013, and whe relevant 2020) as a supplementary Table 2 in the revised R2 version.

“The number of patients for each individual histological subtype of sarcoma or TIM per year, from 2013 to 2016, is therefore presented in these tables. To facilitate the comparison with other databases using previous classifications, the incidence for groups of tumors are also presented in the tables, when they are clinically relevant (e.g. uterine sarcoma). We also considered that it was still useful to present each individual histotype, (e.g. WDLPS) and groups of histotypes when clinically relevant (e.g. WD and DDLPS, or even liposarcoma, leiomyosarcomas). Conversely, when a grouping was not clinically meaningful in clinical routine (e.g. “fibroblastic and myofibroblastic tumors” in table 1) we did not consider this a distinct entity.” 

This is still unclear. The authors should clarify what they count as a single entity. Please add a column on the left hand site of Table 1 and Table 2 indicating what was counted as single entity (N=150). It is not clear also why some grouping have the incidence and others not. it does not seem to depend on the clinical relevance because fibroblastic and myofibroblastic tumours have incidence data in Table 1. Kindly clarify.

Answer : We agree that it is important to clarify the grouping. We have therefore added a supplementary Table 2 indicating all the groups which were used in the present report to facilitate cthe comparision with subsequent data sets.

Incidence

It is a pity not be able to separate TIM from malignant sarcomas. The authors used the WHO classifications which provides information about the sarcoma behaviour. Thus, the authors should clarify why it is not possible to distinguish TIM and malignant sarcomas. This goes back to the issues of understanding the codes used for the analyses.

Answer : To distinguish better TIM from sarcomas, we have added a colour code (blue for TIM in table 4) in the revised version. We hope that this will facilitate the lecture. 

Incidence of individual histotypes and published clinical trials.

“..... 14 of 35 (40%) histotypes with an incidence >1/10 6 had a dedicated phase III study vs 6 of 130 (4.6%) histotypes for sarcomas with a incidence <1/10 6 (p<10 -6 ). 20100 (79,7%) patients of the database had a specific histotype for which no phase III trial had been reported. 21 of 35 (60%) histotypes with an incidence >1/10 6 had a dedicated randomized phase II study vs 10 of 129 (7.7%) histotypes for sarcomas with a incidence <1/10 6 (p<10 -10 ))” 

The authors should clarify whether the histotypes for sarcomas with a incidence <1/106 are 129 or 130.

Answer : We thank the reviewer for the careful lecture. This mistale was corrected , the right number is N=130. This has been modified accordingly in the revised version R2.

Table 1. please check the incidence for Embryonal RMS and Myoepithelioma, myoepithelial carcinoma, & mixed tumour. The incidence of the histotype sum up differently from the incidence reported.

Answer : the sums were wrong and were corrected in the R2 version. We thank the reviewer for a careful attention

Reviewer #2: Review Comments to the Author

Please use the space provided to explain your answers to the questions above. You may also include additional comments for the author, including concerns about dual publication, research ethics, or publication ethics. (Please upload your review as an attachment if it exceeds 20,000 characters) (Limit 100 to 20000 Characters)

Please see my comments in the attached file

Answers : Copied hereunder with the answers.

Major point

Authors wrote: “We investigated then the variability of the yearly incidence of these different tumors in the database. The analysis of variance of the observed incidence indicated a significant interaction between time and histology”. 

However, the analysis of variance is not described in Statistical Analyses

Answer : Indeed, this was not enough described. The analysis of variance was added in the method section ?

Minor points

Where It is I would have written instead

Abstract The 2 later incidence groups The 2 latter incidence groups

Introduction INCa INCA

 … may not treated according to … may not be treated according to 

NETSARC+ network French National Cancer institute French National Cancer Institute

RREPS/NETSARC Database MDTB Acronym never defined

Results This later group a group is not This latter group is not

Anwers : we thank the reviewer for his careful attention. The different points were corrected in the revised version. 

Reviewer #4: The revised version addressed the points risen.

This work will represent the benchmark for future epidemiological studies.

Answer : We thank the reviewer for these kind words

---

## [Decision Letter · Decision Letter 2]

6 Jan 2021

PONE-D-20-18848R2

Nationwide incidence of sarcomas and connective tissue tumors of intermediate malignancy over four years using an expert pathology review network.

PLOS ONE

Dear Dr. Blay,

Thank you for submitting your manuscript to PLOS ONE. After careful consideration, we feel that it has merit but does not fully meet PLOS ONE’s publication criteria as it currently stands. Therefore, we invite you to submit a revised version of the manuscript to improve Engliish writing as a reviewer pointed during the review process.

We look forward to receiving your revised manuscript.

Kind regards,

Sandro Pasquali, M.D., Ph.D.

Academic Editor

PLOS ONE

Additional Editor Comments (if provided):

Following comments from reviewers and authors changes, the manuscript s to be considered accepted, although a revision of the English writing is needed before this manuscript can be considered fully accepted in PLOS ONE.

Reviewers' comments:

Reviewer's Responses to Questions

**Comments to the Author**

1. If the authors have adequately addressed your comments raised in a previous round of review and you feel that this manuscript is now acceptable for publication, you may indicate that here to bypass the “Comments to the Author” section, enter your conflict of interest statement in the “Confidential to Editor” section, and submit your "Accept" recommendation.

Reviewer #1: All comments have been addressed

2. Is the manuscript technically sound, and do the data support the conclusions?

Reviewer #1: Yes

3. Has the statistical analysis been performed appropriately and rigorously? 

Reviewer #1: Yes

4. Have the authors made all data underlying the findings in their manuscript fully available?

Reviewer #1: Yes

5. Is the manuscript presented in an intelligible fashion and written in standard English?

Reviewer #1: Yes

6. Review Comments to the Author

Reviewer #1: thank you for the revised version of the paper. I would recommend an english revision of the text. the different revision improved the understanding but in some paragraphs need english edits.

7. PLOS authors have the option to publish the peer review history of their article (what does this mean?). If published, this will include your full peer review and any attached files.

Reviewer #1: No

---

## [Author Response · Author response to Decision Letter 2]

13 Jan 2021

JY Blay

Jean-yves.blay@lyon.unicancer.fr

to

Sandro Pasquali, M.D., Ph.D.

Academic Editor

PLOS ONE

PONE-D-20-18848R1 Nationwide incidence of sarcomas and connective tissue tumors of intermediate malignancy over four years using an expert pathology review network.

Dear Dr. Pasquali,

Thank you for your recent e-mail regarding the above mentioned article submitted to PLOS ONE. 

We are now able to submit a revised version of this article after a review by native English speakers we have included in the acknowledgement section. 

A revised version with visible marks in included as well as the revised version. 

We remain of course at your disposition for any further question you may have regarding the work

With kind regards

 JY Blay

---

## [Editor Report · Decision Letter 3]

29 Jan 2021

Nationwide incidence of sarcomas and connective tissue tumors of intermediate malignancy over four years using an expert pathology review network.

PONE-D-20-18848R3

Dear Dr. Blay,

We’re pleased to inform you that your manuscript, after revision of the English writing, has been judged scientifically suitable for publication and will be formally accepted for publication once it meets all outstanding technical requirements.

Kind regards,

Sandro Pasquali, M.D., Ph.D.

Academic Editor

PLOS ONE

---

## [Editor Report · Acceptance letter]

11 Feb 2021

PONE-D-20-18848R3 

Nationwide incidence of sarcomas and connective tissue tumors of intermediate malignancy over four years using an expert pathology review network. 

Dear Dr. Blay:

I'm pleased to inform you that your manuscript has been deemed suitable for publication in PLOS ONE. Congratulations! Your manuscript is now with our production department. 

Kind regards, 

on behalf of

Dr. Sandro Pasquali 

Academic Editor

PLOS ONE